# Value construction through sequential sampling explains serial dependencies in decision making

**Ariel Zylberberg[1]\*, Akram Bakkour[1,2,3], Daphna Shohamy[1,4,5†],
Michael N Shadlen[1,4,6,7†]**

[1]Mortimer B Zuckerman Mind Brain Behavior Institute, Columbia University, New York, United States; [2]Department of Psychology, University of Chicago, Chicago, United States; [3]Neuroscience Institute, University of Chicago, Chicago, United States; [4]Department of Neuroscience, Columbia University, New York, United States; [5]Department of Psychology, Columbia University, New York, United States; [6]The Kavli Institute for Brain Science, Columbia University, New York, United States; [7]Howard Hughes Medical Institute, Chevy Chase, United States

\*For correspondence:
ariel.zylberberg@gmail.com

†These authors contributed equally to this work

Competing interest: The authors declare that no competing interests exist.

## eLife Assessment

This **important** study addresses key assumptions underlying current models of the formation of value-based decisions. The authors provide **convincing** evidence that the subjective values human participants assign to items change across sequences of multiple decisions. They establish methods to detect these changes in frequently used behavioral task designs.

**Abstract** Deciding between a pair of familiar items is thought to rely on a comparison of their subjective values. When the values are similar, decisions take longer, and the choice may be inconsistent with stated value. These regularities are thought to be explained by the same mechanism of noisy evidence accumulation that leads to perceptual errors under conditions of low signal to noise. However, unlike perceptual decisions, subjective values may vary with internal states (e.g. desires, priorities) that change over time. This raises the possibility that the apparent stochasticity of choice reflects changes in value rather than mere noise. We hypothesized that these changes would manifest in serial dependencies across decision sequences. We analyzed data from a task in which participants chose between snack items. We developed an algorithm, *Reval*, that revealed significant fluctuations of the subjective values of items within an experimental session. The dynamic values predicted choices and response times more accurately than stated values. The dynamic values also furnished a superior account of the BOLD signal in ventromedial prefrontal cortex. A novel bounded-evidence accumulation model with temporally correlated evidence samples supports the idea that revaluation reflects the dynamic construction of subjective value during deliberation, which in turn influences subsequent decisions.

## Introduction

A central idea in decision theory and economics is that each good can be assigned a scalar utility value that reflects its desirability. The concept of utility, or subjective value, provides a common currency for comparing dissimilar goods (e.g. pears and apples) such that decision-making can be reduced to estimating the utility of each good and comparing them (*von Neumann and Morgenstern, 1944*;

*Samuelson, 1937*; *Montague and Berns, 2002*). The idea is supported by studies that have identified neurons that correlate with the subjective value of alternatives in various brain structures, most notably the ventromedial prefrontal cortex, and it is so pervasive that decisions based on preferences are often referred to as 'value-based decisions' (*Kable and Glimcher, 2007*; *Kim et al., 2008*; *Padoa-Schioppa and Assad, 2006*).

Choice and response time (RT) in simple perceptual and mnemonic decisions are often modeled within the framework of bounded evidence accumulation (BEA). The framework posits that evidence samples for and against the different options are accumulated over time until the accumulated evidence for one of the options reaches a threshold or bound (*Ratcliff, 1978*; *Gold and Shadlen, 2007*). A case in point is the random dot motion (RDM) discrimination task, in which participants must decide whether randomly moving dots have net rightward or leftward motion, while the experimenter controls the proportion of dots moving coherently in one direction, termed the *motion strength* (e.g. *Gold and Shadlen, 2007*). BEA models explain the choice, RT, and confidence in the RDM task under the assumption that the rate of accumulation, often termed the *drift rate*, depends on motion strength (*van den Berg et al., 2016*; *Zylberberg and Shadlen, 2024*). Value-based decisions have also been modeled within the framework of BEA. The key assumption is that at any given time, decision-makers only have access to a noisy representation of the subjective value of each item, and the drift rate depends on the difference between the subjective values of the items (*Krajbich et al., 2010*; *Thomas et al., 2019*; *Sepulveda et al., 2020*; *Bakkour et al., 2019*).

A condition that renders the BEA framework normative is that the noise corrupting the evidence samples is independent, or equivalently, that the evidence samples are conditionally independent given the drift rate. For example, in modeling the RDM and other perceptual decision making tasks, evidence samples are assumed to be independent of each other, conditioned on motion strength and direction (e.g. *Zylberberg et al., 2016*). This assumption is sensible because (*i*) the main source of stochasticity in perceptual decision making is the noise affecting the sensory representation of the evidence, which has a short-lived autocorrelation, and (*ii*) these decisions are often based on an evidence stream (e.g. a dynamic random dot display) that provides conditionally independent samples, by design. The assumption of conditional independence justifies the process of evidence accumulation, because accumulation (or averaging) can only remove the noise components that are not shared by the evidence samples.

For value-based decisions, the assumption of conditional independence is questionable. Alternatives often differ across multiple attributes (e.g. *Busemeyer and Townsend, 1993*; *Tversky, 1977*). For example, when choosing between different snacks, they may differ in calories, healthiness, palatability, and so on (*Suzuki et al., 2017*). The weight given to each attribute depends on the decision-maker's internal state (*Noguchi and Stewart, 2018*; *Juechems and Summerfield, 2019*). This internal state includes desires, needs, priorities, attentional state and goals. We use the term mindset, or state of mind, to refer to all these internal influences on valuation. A mindset can be persistent. For example, a famished decision-maker may prioritize the nutritional content of each food when making a choice. Under less pressing circumstances, the salience of an attribute may be suggested by snack alternatives themselves. For example, seeing French fries may make us aware that we crave something salty, and saltiness becomes a relevant attribute informing the current decision and possibly future decisions too. The examples illustrate how a decision-maker's mindset can shift rapidly or meander, based on the attributes in focus or the identity of the items under consideration (*Shadlen and Shohamy, 2016*; *Stewart et al., 2006*). Importantly, mindset is dynamic. It can change abruptly, motivated by a thought in an earlier trial or by interoception during deliberation (e.g. thirst). Unlike perceptual decision-making, where the expectation of a sample of evidence is thought to be fixed, conditional on the stimulus, the expectation of the evidence bearing on preference is itself potentially dynamic.

We sought to test the notion that the desirability of an item changes as a result of the deliberation that leads to a choice. We hypothesized that if subjective values are dynamic, then value-based decisions should exhibit serial dependencies when multiple decisions are made in a sequence. A choice provides information not only about which option is preferred, but also about the decision maker's mindset at the moment of the choice (e.g. whether they prioritize satiation or palatability). Therefore, a choice is informative about future choices because the decision maker's *mindset* is likely to endure longer than a single decision, or even multiple decisions.

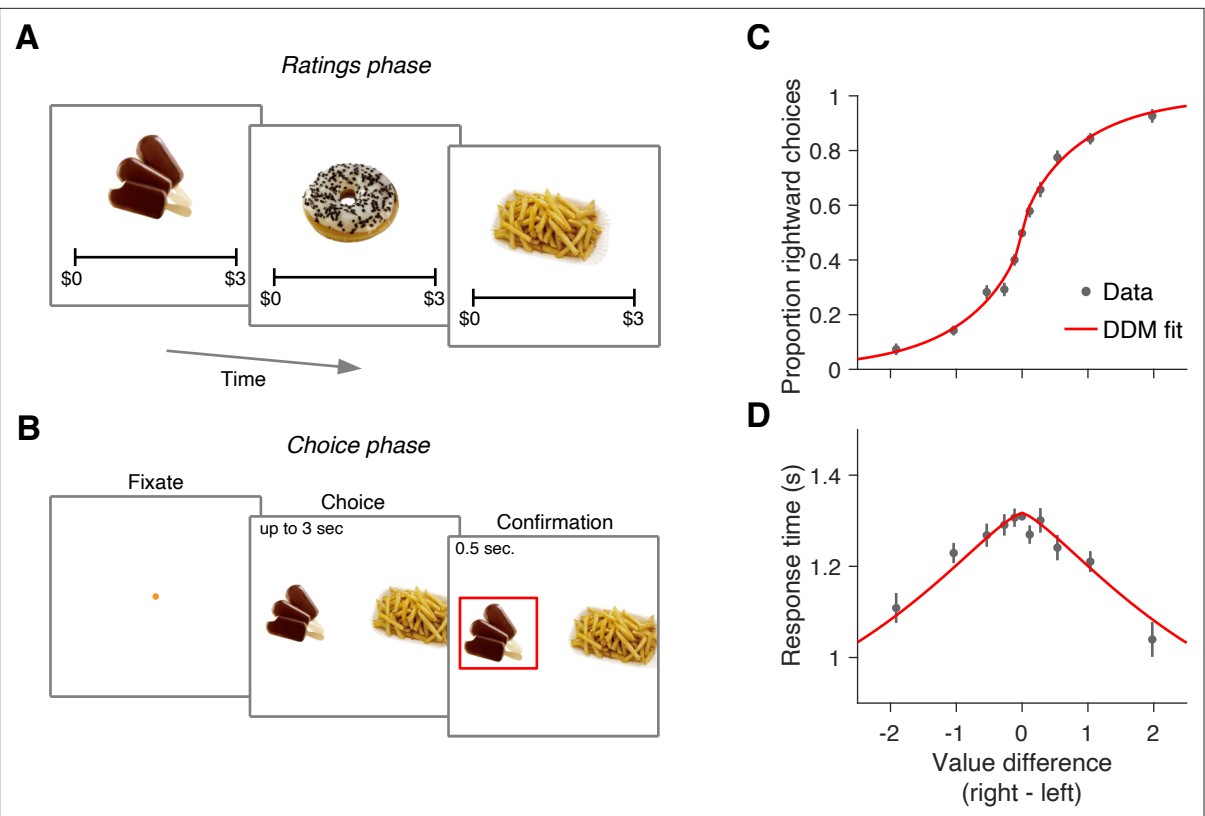

**Figure 1.** Food choice task. (**A**) In an initial 'ratings' task, participants were shown 60 individual appetizing snack items and asked to indicate how much they would be willing to pay for each item using a monetary scale ranging from $0 to $3. (**B**) In the main experiment, participants were presented with pairs of snack items and asked to choose which one they would prefer to consume at the end of the session. After making their choice, the chosen item was highlighted by a square box for an additional 0.5 s. Each of the 30 participants completed 210 trials, with each item appearing seven times during the experiment. A subset of 60 item pairs were repeated once. (**C**) Proportion of trials in which participants selected the right item as a function of the difference in value between the right and left items ($\Delta v_s$). Proportions were first determined for each participant and then averaged across participants. Error bars indicate the s.e.m. across participants. (**D**) Mean response time as a function of the difference in value between the right and left items. Error bars indicate the s.e.m. across participants. Red curves in panels C-D are fits of a drift-diffusion model (DDM).

We reanalyzed data from *Bakkour et al., 2019*. Participants were presented with pairs of snacks and had to choose the one they preferred. This *Food choice task* has been used extensively to study the sequential sampling process underlying value-based decisions (e.g. *Krajbich et al., 2010*). Crucially, in the *Bakkour et al., 2019* experiment, each item was presented multiple times, allowing us to infer how preference for an item changes during a single experimental session. Using a novel algorithm we call *Reval*, we show that the subjective value of items changed over the session. The revaluation was replicated in a sequential sampling model in which successive samples of evidence are not assumed to be conditionally independent. We argue that the revaluation process we observed reflects a process by which the value of the alternatives is constructed during deliberation by querying memory and prospecting for evidence that bears on desirability (*Lichtenstein and Slovic, 2006*; *Johnson et al., 2007*).

## Results

### Food choice task

We re-examined data from a previous study in which 30 participants completed a food choice task (*Bakkour et al., 2019*). Prior to the main experiment, participants were asked to indicate their willingness to pay for each of 60 snack items on a scale from 0 to US$3 (*Figure 1A*). We refer to these explicitly reported values as *s-values*, or $v_s$ (where $s$ stands for 'static' as opposed to the 'dynamic' values we define below). In the main experiment (conducted in an MRI scanner), participants were shown pairs of

images of previously rated snack items and had to choose which snack they would prefer to consume at the end of the study (*Figure 1B*).

The data from *Bakkour et al., 2019* replicate the behavior typically observed in such tasks. Both choice and response time were systematically related to the difference in *s-value*, $\Delta v_s$, between the right and left items. Participants were more likely to choose the item to which they assigned a higher value during the rating phase (p<0.0001; $\mathcal{H}_0 : \beta_1 = 0$; *Equation 2*). They were also more likely to respond faster when the absolute value of the difference between the items, $|\Delta v_s|$, was greater (p<0.0001; $\mathcal{H}_0 : \beta_1 = 0$; *Equation 3*).

The relationship between $\Delta v_s$, choice, and response time is well described by a bounded evidence accumulation model (*Krajbich et al., 2010*; *Bakkour et al., 2019*). The solid lines in *Figure 1C–D* illustrate the fit of such a model in which the drift rate depends on $\Delta v_s$. Overall, the behavior of our participants in the task is similar to that observed in other studies using the same task (e.g. *Krajbich et al., 2010*; *Folke et al., 2016*; *Sepulveda et al., 2020*).

### Limited power of explicit reports of value to explain binary choices

An intriguing aspect of the decision process in the food choice task is its highly stochastic nature. This is evident from the shallowness of the choice function (*Figure 1C*): participants chose the item with a higher *s-value* in only 64% of the trials. This variability is typically attributed to unspecified noise when recalling item values from memory (e.g. *Krajbich et al., 2010*). An alternative explanation is rooted in constructive value theories, which suggest that the value of each item is constructed, not retrieved, during the decision process (*Lichtenstein and Slovic, 2006*; *Shadlen and Shohamy, 2016*; *Johnson et al., 2007*). This construction process is sensitive to the context in which it is elicited (e.g. the identity

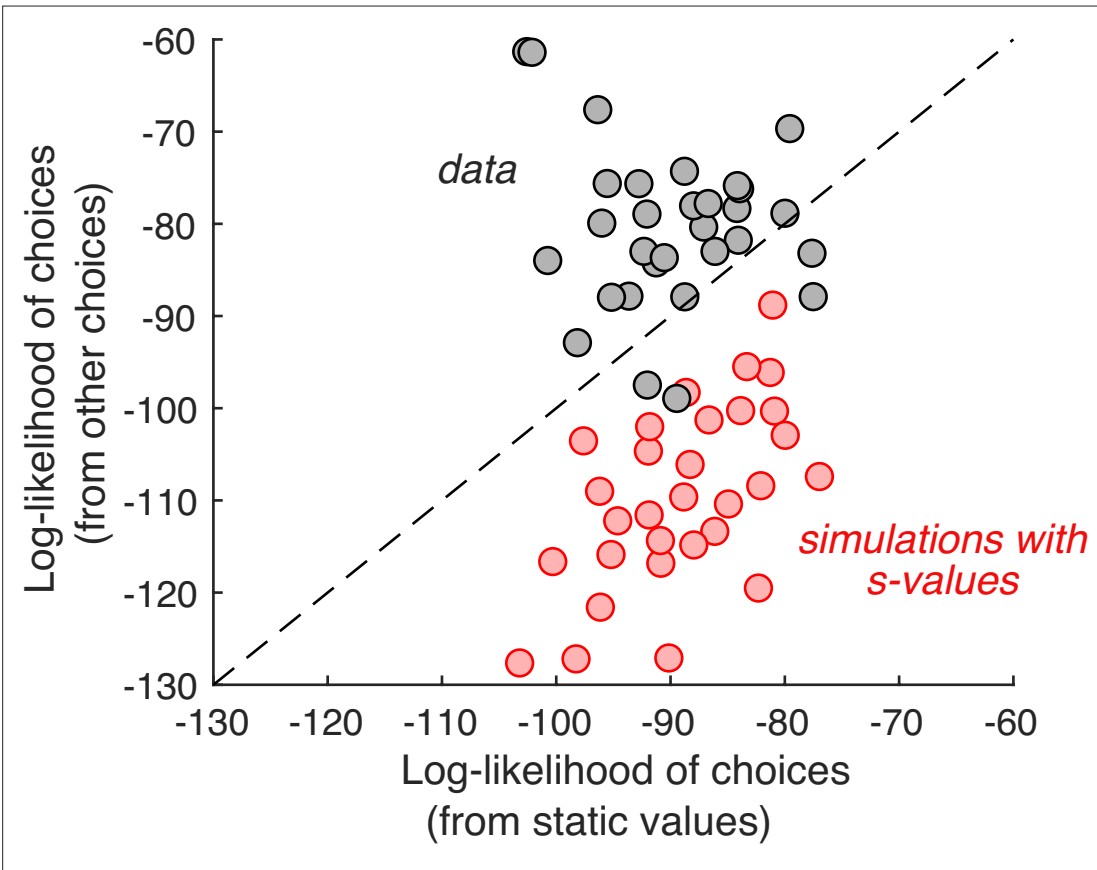

**Figure 2.** Individual choices are better explained by values inferred from the other trials than values reported in the ratings task. Gray data points represent the total log-likelihood of each participant's choices, given two types of predictions: (*abscissa*) from a logistic regression, fit to the static values; (*ordinate*) from a procedure that infers the values based on choices on the other trials. Predictions derived from the other trials are better in all but four participants. The red markers were obtained using the same procedure, applied to choices simulated under the assumption that the *s-values* are the true values of the items. It shows that the inferential procedure is not guaranteed to improve predictions.

of items being compared), so the values reported during the valuation process may differ from those used in the choice task. According to this idea, the apparently stochastic choice is a veridical reflection of the constructed values.

If this were true, then the choice on any one *cynosure* trial—that is, the trial we are scrutinizing—would be better explained by values inferred from the choices on the other trials than by the *s-values*. We therefore compared two regression models that produce the log odds of the choice on each *cynosure* trial. The first regression model uses the *s-values* plus a potential bias for the left or right item. The second regression model includes one regression coefficient per item plus a left/right bias. It uses all the other trials (except repetitions of the identical pair of items) to establish the weights. While this model has more free parameters, the comparison is valid because we are using the models to predict the choices made on trials that were not used for model fitting. The better model is the one that produces larger log odds of the choice on the *cynosure* trial. As shown in *Figure 2*, the second regression model is superior.

To ensure that this result is not produced artifactually from the algorithm, we performed the same analysis on simulated data. We fit the experimentally observed choices using a logistic regression model with $\Delta v_s$ and an offset as independent variables, and simulated the choices by sampling from Bernoulli distributions with parameter, $p$, specified by the logistic function that best fit each participant's choices (i.e., weighted-coin flips). We repeated the model comparison using the simulated choices and found that, contrary to what we observed in the experimental data, the model using explicit value reports is the better predictor (*Figure 2*, red).

Taken together, these analyses show that explicit value reports have limited power to predict choices, which partially explains their apparent stochasticity (*Konovalov and Krajbich, 2019*; *Verhoef and Franses, 2003*; *Wardman, 1988*). In the following sections, we elaborate on this observation. Not only do the values used to make the binary choices differ from the *s-values*, they drift apart during the experiment. We show that these changes arise through the deliberative process leading to the preference decisions themselves.

## Preferences change over the course of the experiment

In the experiment, a subset of snack pairs were presented twice, in a random order within the sequence of trials. These trials allow us to assess whether preferences change over the course of a session. For

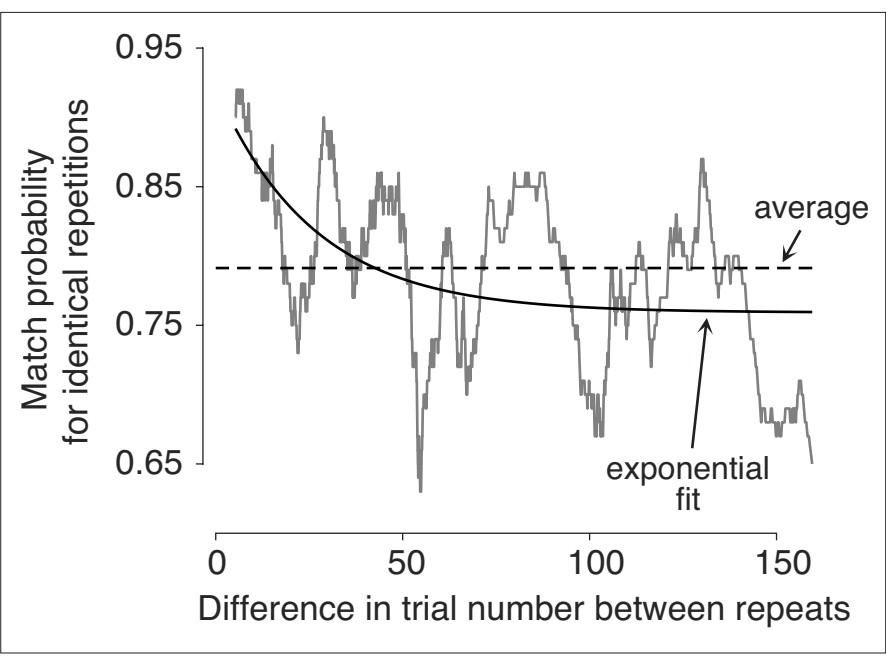

**Figure 3.** Preferences change over time. Probability of making the same choice on the two trials with the same item pair, shown as a function of the difference in trial number between them ($\Delta$tr). Trial pairs with identical items (N=1726) were sorted by $\Delta$tr, and the match probabilities were smoothed with a boxcar function with a width of 100 observations.

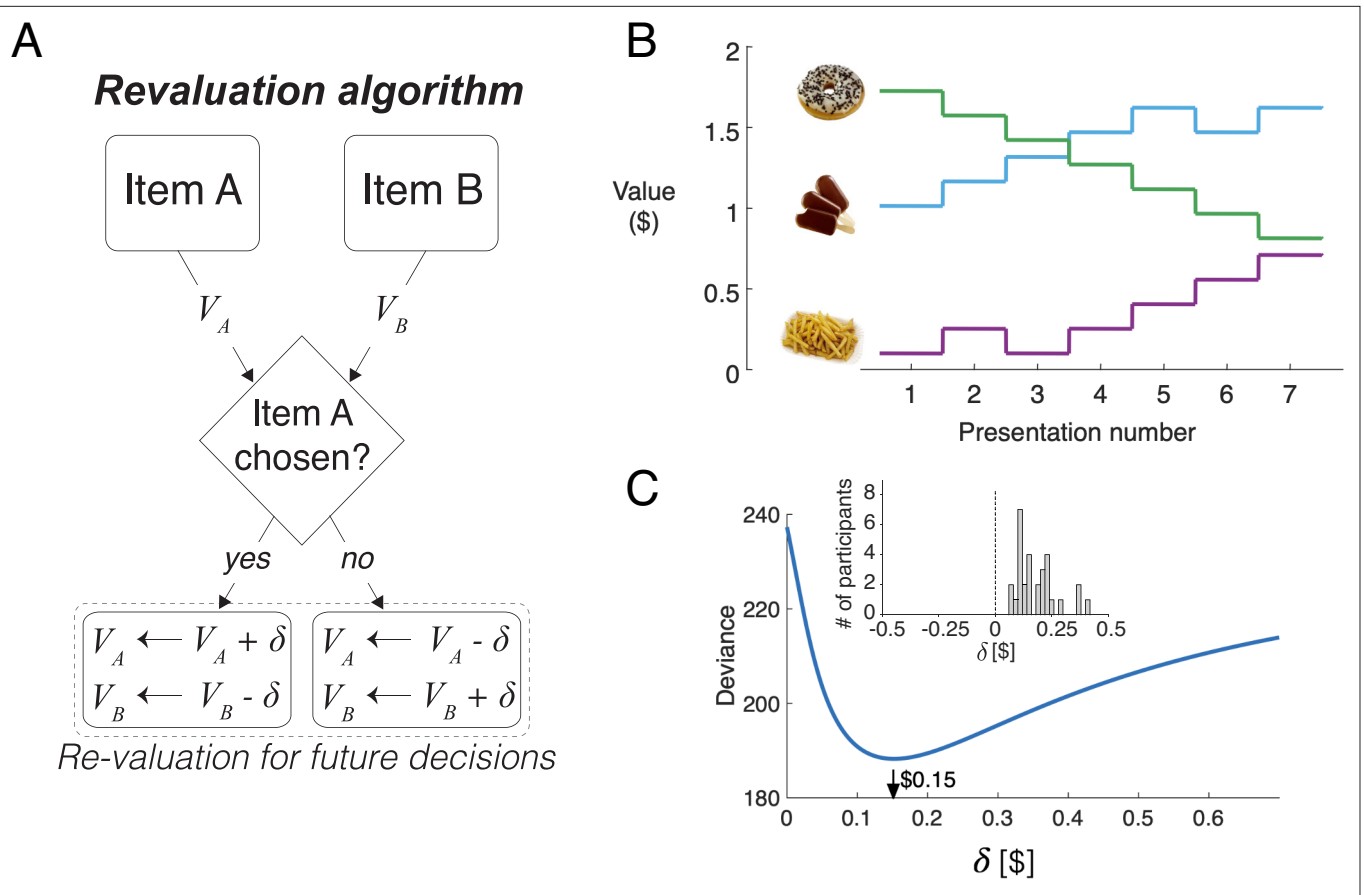

**Figure 4.** Revaluation algorithm. (**A**) Schematic example of the revaluation algorithm applied to one decision. After a choice between items A and B, the value of the chosen item is increased by $\delta$ and the value of the unchosen item is decreased by the same amount. (**B**) Example of value changes due to revaluation, for three items, as a function of the presentation number within the session. In the experiment, each item was presented seven times. (**C**) Deviance of the of the logistic regression model fit with the values assigned by the Reval procedure to the data from one participant, for different values of $\delta$. The best fitting value is $0.15. The inset shows a histogram of the best-fitting $\delta$ values across participants.

these duplicated item pairs, we calculate the average number of times that the same item was chosen on both presentations—which we refer to as the *match probability*. Participants were more likely to select the same option when presentations of the same pair were closer in time (*Figure 3*). To assess the significance of this effect, we fit a logistic regression model using all pairs of trials with identical stimuli to predict the probability that the same item would be chosen on both occasions. The regression coefficient associated with the number of trials between repetitions was negative and highly significant (p<0.0001; t-test, *Equation 8*). It therefore follows that preferences are not fixed, not even over the course of a single experimental session.

## Choice alternatives undergo revaluation

We propose a simple algorithm to characterize how preferences changed over the course of the session. It assumes that on each decision, the value of the chosen item increases by an amount equal to $\delta$, and the value of the unchosen item decreases by the same amount (*Figure 4A*). We refer to the updated values as *d-values*, or $v_d$, where $d$ stands for 'dynamic'.

*Figure 4B* illustrates how the value of the items changes over the course of the session, for a given value of $\delta$, for three snack items. For example, while the item shown with the green curve is initially very valuable, as indicated by its high initial rating, its value decreases over the course of the session each time it was not selected.

We determined the degree of revaluation that best explained the participants' choices. For each participant, we find the value of $\delta$ that minimizes the deviance of a logistic regression model that uses the *d-values* to fit the choices made on each trial,

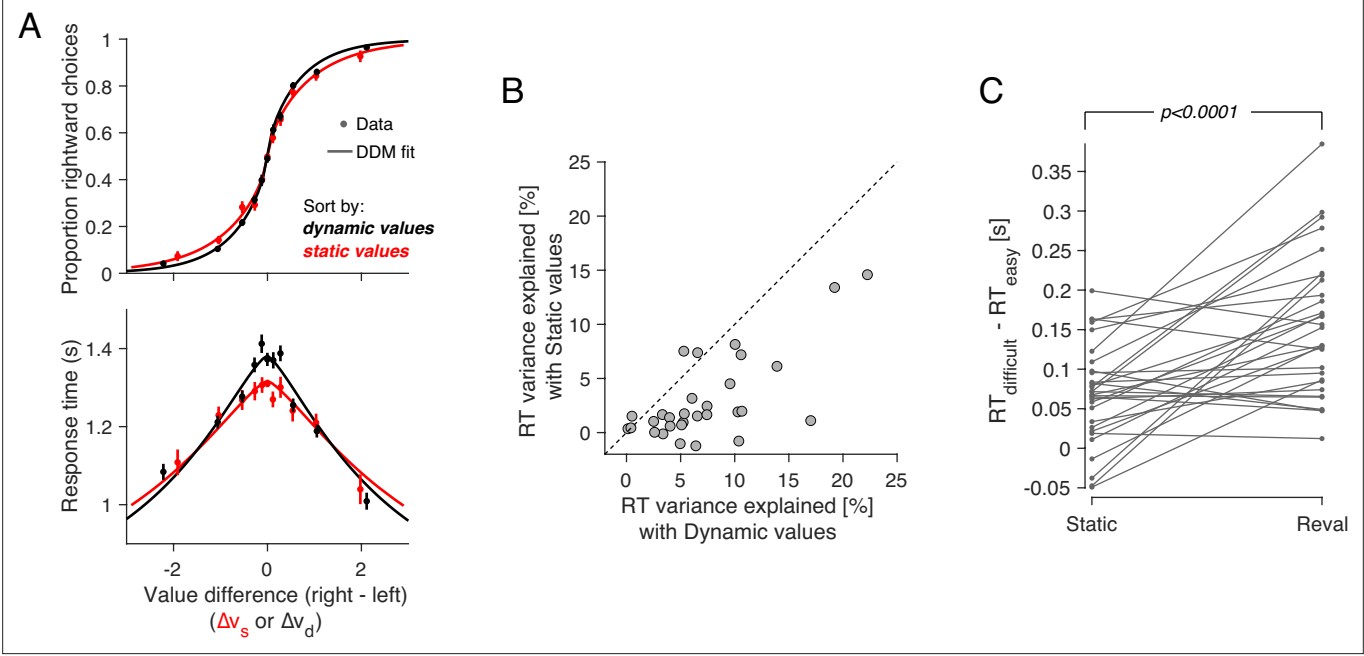

**Figure 5.** Revaluation explains choice and RT better than static values. (**A**) Proportion of rightward choices (top) and mean response time (bottom) as function of the difference in *d-value* between the two items. The black lines are fits of a drift-diffusion model that uses the *d-values*. The red lines correspond to the fits of a DDM that uses the *s-values* (same as in **Figure 1C–D**). Error bars indicate s.e.m. across trials. Participants are more sensitive to *d-values* than *s-values* (top) and the *d-values* better explain the full range of RTs (bottom). (**B**) Percentage of variance in response times explained by a *DDM* in which the drift rate depends on either $\Delta v_d$ (abscissa) or $\Delta v_s$ (ordinate). Each data point corresponds to a different participant. For most participants, the model based on the dynamic values explained a greater proportion of the variance. (**C**) *d-values* are better than *s-values* at predicting the difficulty of a decision as reflected in the response times. Data points represent the difference in mean RTs between difficult and easy decisions. Positive values indicate that difficult decisions take longer on average than easy ones. *Difficult* and *easy* are defined relative to the median of the absolute value of $\Delta v_s$ (left) or $\Delta v_d$ (right). The lines connect the mean RTs of each participant. P-value is from a paired t-test.

The online version of this article includes the following figure supplement(s) for figure 5:

**Figure supplement 1.** Static and dynamic values competing to explain choice.

**Figure supplement 2.** Comparison of *DDM* fits using static and dynamic values.

**Figure supplement 3.** Similar $\delta$ values obtained by *Reval* and logistic regression.

$$\text{logit}[p_{\text{choice}}] = \beta_0 + \beta_1 v_d^{(\text{left})} + \beta_2 v_d^{(\text{right})}, \tag{1}$$

where $p_{\text{choice}}$ is the probability of choosing the item that was presented on the right. The *d-values* are initialized to the explicitly reported values for all items, and they are updated by plus or minus $\delta$ when an item is chosen or rejected, respectively. Importantly, the updated values only affect future decisions involving the items.

*Figure 4C* shows the deviance of the logistic regression model for a representative participant, as a function of $\delta$. For this participant, the best explanation of the choices is obtained with a value of $\delta \approx \$0.15$. We fit the value of $\delta$ independently for each participant to minimize the deviance of the logistic regression model fit to the choices. On average, each choice changed the value of the chosen and unchosen items by $\$0.18 \pm 0.016$ (mean ± s.e.m., (*Figure 4C*), inset).

The values derived from the *Reval* algorithm explain the choices better than the explicit value reports. The choices are more sensitive to variation in $\Delta v_d$, evidenced by the steeper slope (*Figure 5A*). When $\Delta v_d$ and $\Delta v_s$ are allowed to compete for the same binomial variance, the former explains away the latter. This assertion is supported by a logistic regression model that incorporates both $\Delta v_s$ and $\Delta v_d$ as explanatory variables (*Equation 7*). The coefficient associated with $\Delta v_s$ is not significantly different from zero while the one associated with $\Delta v_d$ remains positive and highly significant (*Figure 5—figure supplement 1*).

More surprisingly, *Reval* allows us to explain the response times better than the explicit value reports, even though RTs were not used to establish the *d-values*. We used the *d-values* to fit a

drift-diffusion model to the single-trial choice and response time data, and compared this model with the one that was fit using the *s-values* (*Figure 5A*). To calculate the fraction of RT variance explained by each model, we subtracted from each trial's RT the models' expectation, conditional on $\Delta v_x$ (with $x \in \{s, d\}$) and choice. The model that relies on the *d-values* explains a larger fraction of variance in RT than the model that relies on the *s-values* (*Figure 5B*). This indicates that the re-assignment of values following *Reval* improved the capacity of a *DDM* to explain the response times.

The *DDM* that uses the dynamic values also explains the combined choice-RT data better than the one that uses the static values. We compared their goodness of fit using the Bayesian Information Criteria (BIC), penalizing the *DDM* that uses the dynamic values for the revaluation update parameter, $\delta$. For all participants, the *DDM* that uses the dynamic values provided a better fit than the *DDM* that uses the static values (*Figure 5—figure supplement 2A*). To control for the possibility that the model comparison is biased by the extra parameter in the dynamic model ($\delta$), we simulated choice and RT data for each participant from the *DDM* model fit to the static values, and fit these simulated data to the DDMs using static and dynamic values (in the latter case applying the *Reval* algorithm prior to fitting). For the simulated data, the model comparison favored the *DDM* using static values for most participants (*Figure 5—figure supplement 2B*), indicating that the additional parameter in the dynamic model does not strongly bias the model comparison.

The time it takes to make a decision, and the difference in value between the items under comparison, can be considered complementary measures of decision difficulty. On average, the more similar in value the two items are, the longer it would take to commit to a choice. Under this assumption, we can compare how well the static and the dynamic values predict the difficulty of the choices as judged by their response times. The application of *Reval* revealed that some decisions that were initially considered difficult, because $\Delta v_s$ was small, were actually easy, because $\Delta v_d$ was large, and vice versa. Grouping trials by the $\Delta v_d$ led to a wider range of mean RTs compared to when we grouped them by $\Delta v_s$ (*Figure 5A*). The effect can also be observed for individual participants (*Figure 5C*). For each participant, we grouped trials into two categories depending on whether the difference in value was less than or greater than the median difference. We then calculated the mean RT for each of the two groups of trials. The difference in RT between the two groups was greater when we grouped the trials using the *d-values* than when we used the *s-values*. This implies the *d-values* were better than the *s-values* at assessing the difficulty of a decision as reflected in the response time.

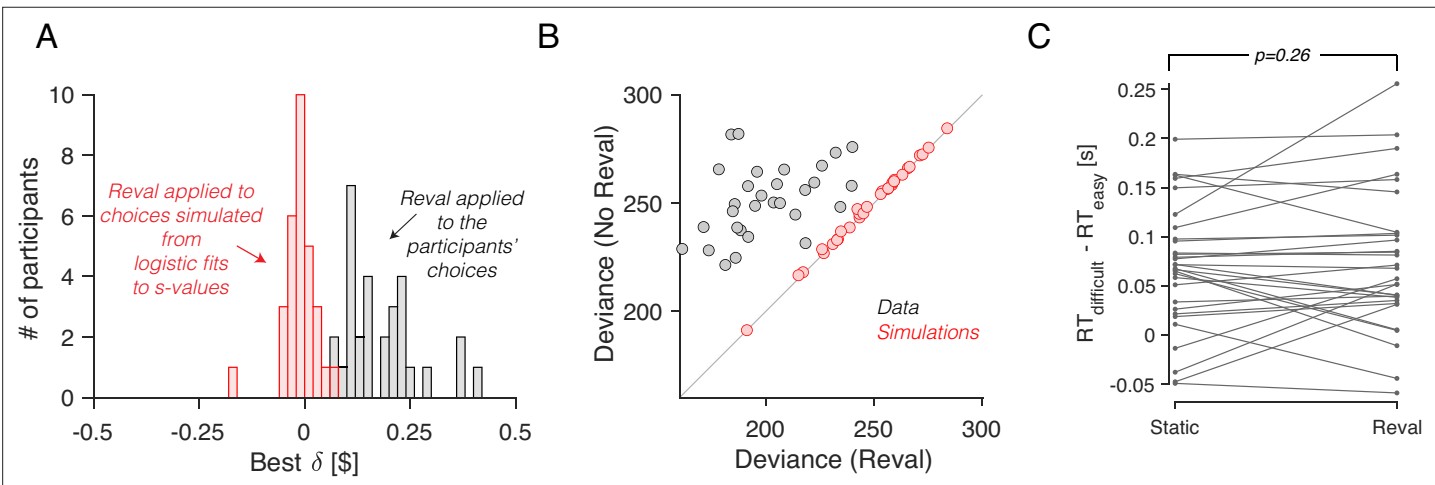

**Figure 6.** No revaluation in simulated data. (**A**) Histogram of the best-fitting revaluation update ($\delta$) for data simulated by sampling choices from a logistic function fit to the participants' choices. The best-fitting $\delta$ values for the simulated choices are centered around 0. For reference, we have also included a histogram of the $\delta$ values obtained from the fits to the participants' data, showing all positive values (gray). (**B**) Deviance of the logistic regression model used to explain the choices (*Equation 1*), fit using either the static values (ordinate) or the *Reval* algorithm (abscissa). Each data point corresponds to a different participant. Experimental data are shown in gray and simulated data (as in panel A) are shown in red. The marked reduction in deviance in the experimental data is absent in the data simulated by sampling from logistic regressions fit to the static values. (**C**) The same analysis shown in *Figure 5C*, applied to the simulated data. The values obtained from *Reval* were no better than the static values at explaining the RTs, as expected, since the $\delta$ values were ~0 and thus $v_d \approx v_s$. Same conventions as in *Figure 5C*.

We verified that the improvement in fit was not just due to the additional free parameter ($\delta$). To do this, we again used simulated choices sampled from logistic regression models fit to the participants' choices, as we did for *Figure 2*. Because the choices are sampled from logistic functions fit to the choice data, they lead to a psychometric function that is similar to that obtained with the experimental data. We reasoned that if revaluation were an artifact of the analysis method, then applying the revaluation algorithm to these simulated data should lead to values of $\delta$ and goodness of fit similar to those of the real data. To the contrary, (*i*) the optimal values of $\delta$ for the simulated data were close to zero (*Figure 6A*); (*ii*) the reduction in deviance after applying *Reval* was negligible compared to the reduction in the actual data (*Figure 6B*); and (*iii*) we found no significant difference in the RT median splits between *s-values* and *d-values* (*Figure 6C*). This shows that the improvements in fit quality due to *Reval* are neither guaranteed nor an artifact of the procedure.

## Imperfect value reports do not explain revaluation away

The idea that a choice can induce a change in preference is certainly not new (*Festinger, 1957*). Choice-induced preference change (CIPC) has been documented using a *free-choice paradigm* (*Brehm, 1956*), whereby participants first rate several items, and then choose between pairs of items to which they have assigned the same rating, and finally rate the items again. A robust finding is that items that were chosen are given higher ratings and items that were not chosen are given lower ratings relative to pre-choice ratings, leading to the interpretation that the act of choosing changes the preferences for the items under comparison. However, it has been suggested that the CIPC demonstrated with the free-choice paradigm can be explained as an artifact (*Chen and Risen, 2010*). Put simply, the initial report of value may be a noisy rendering of the true latent value of the item. If two items, A and B, received the same rating but A was chosen over B, then it is likely that the true value for item A is greater than for item B, not because the act of choosing changes preferences, but because the choices are informative about the true values of the items, which are unchanging.

We examined whether *Reval* could be explained by the same artifact. We considered the possibility that the items' valuation in the choice phase are static but potentially different from those reported in the ratings phase. If the values are static, but different from those explicitly reported, then *Reval* could still improve choice and RT predictions by revealing the true subjective value of the items.

We reasoned that if values were static, the improvements we observed in the logistic fits when we applied *Reval* should be the same regardless of how we ordered the trials before applying it. To test

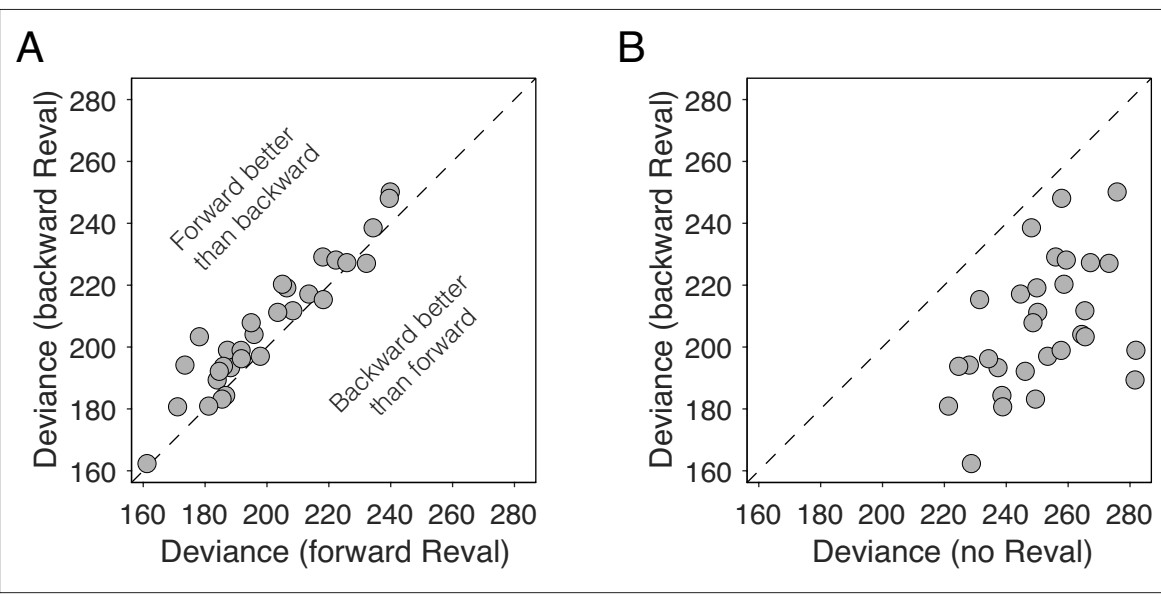

**Figure 7.** *Reval* is sensitive to trial order. (**A**) Deviance obtained by applying *Reval* to the trials in the order in which they were completed (abscissa) and in the reverse order (ordinate). Each data point corresponds to a different participant. The deviance is greater (i.e. the fits are worse) when *Reval* is applied in the reverse direction. (**B**) The deviance of the logistic regression model used to explain the choices (*Equation 1*), obtained by applying *Reval* in the backward direction (ordinate), is lower (i.e. the fits are better) than the deviance obtained using the static values (abscissa). Each data point corresponds to a different participant.

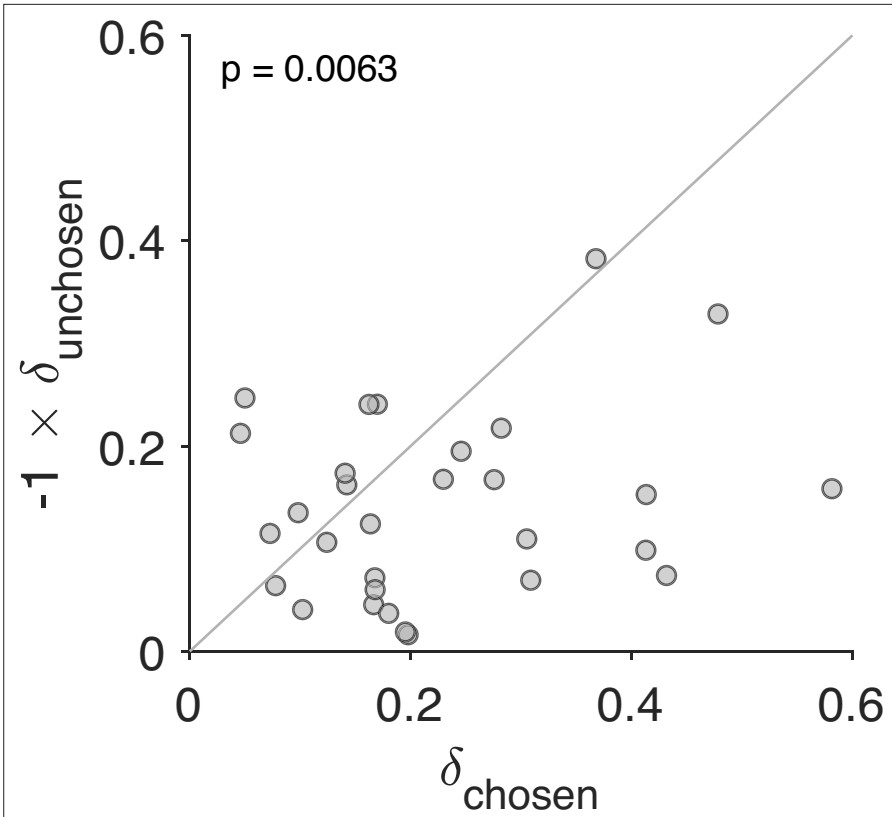

**Figure 8.** Stronger revaluation for the chosen than for the unchosen item. We fit a variant of the *Reval* algorithm that includes separate update values ($\delta$) for the chosen and unchosen options. The best-fitting $\delta$ value for the chosen option (abscissa) is plotted against the best-fitting value for the unchosen option (ordinate). Each data point corresponds to one participant. The increase in value for the chosen option is greater than the decrease in value for the unchosen option (paired t-test).

---

this, we applied *Reval* in the order in which the trials were presented in the experiment, and also in the reverse order (i.e. from the last trial to the first). If the values were static, then the quality of the fits should be statistically identical in both cases. In contrast, we observed that the variance explained by *Reval* was greater (i.e. the deviance was lower) when it was applied in the correct order than when it was applied in the opposite order (*Figure 7A*; p<0.0001, paired t-test). This rules out the possibility that the values were static. Moreover, the values produced by applying *Reval* in the reverse direction explained the choices better than the static values (*Figure 7B*). This might seem counterintuitive, given that the initial values for the *Reval* algorithm are the *s-values*, which are explicitly reported *before* the main experiment. In a later section, we show that this effect stems from the same process that gives rise to revaluation (Is revaluation a byproduct of deliberation?).

## Asymmetric value-updating for chosen and unchosen options

So far we have assumed that a choice increases the value of the chosen option by $\delta$ and decreases the value of the unchosen option by the same amount. Here, we evaluate the possibility that the degree of revaluation is different for the chosen and unchosen options. We fit a variant of the *Reval* algorithm with two values of $\delta$, one for the chosen option ($\delta_{chosen}$) and one for the unchosen option ($\delta_{unchosen}$). *Figure 8* shows the values that best fit the data for each participant. For each participant, $\delta_{chosen}>0$ and $\delta_{unchosen}<0$; in other words, the value of the chosen item typically increases, while the value of the unchosen item tends to decrease following a choice. Further, for most participants, the degree of revaluation is greater for the chosen option than for the unchosen option. As we speculate in the discussion, this result may be related to the unequal distribution of attention between the chosen and unchosen items (*Krajbich et al., 2010*).

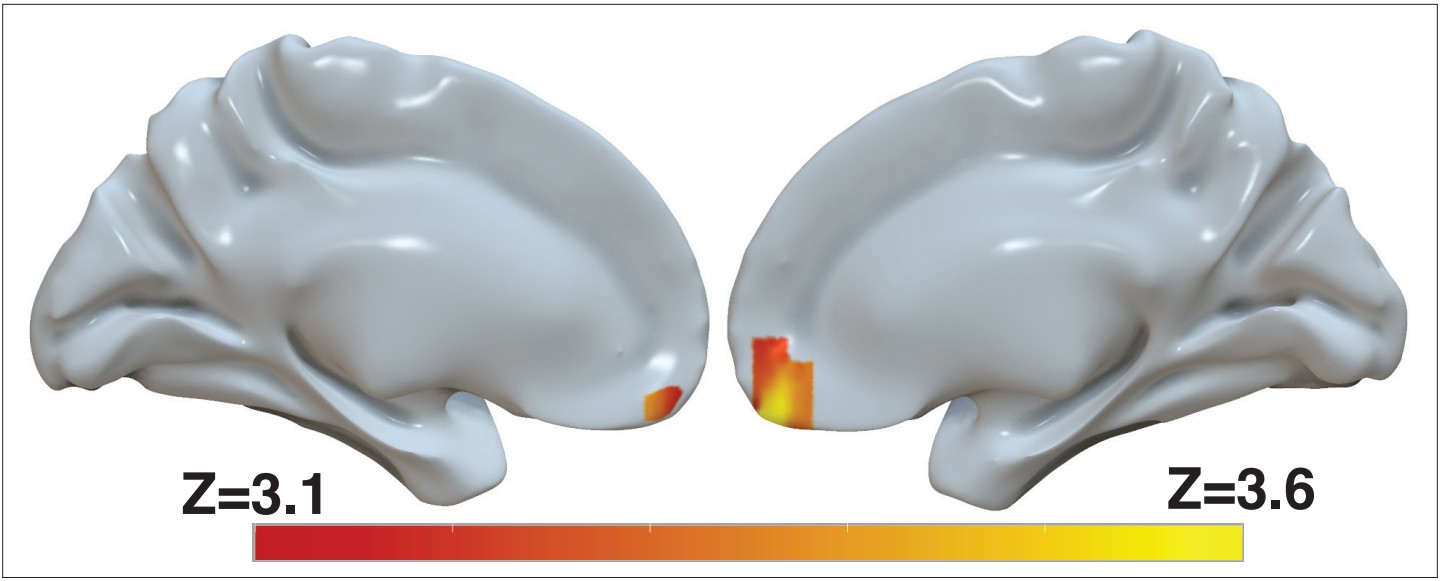

**Figure 9.** Revaluation reflected in BOLD activity in ventromedial prefrontal cortex. Brain-wide fMRI analysis revealed a significant correlation between *d-values* and activity in the vmPFC, after controlling for *s-values*. The statistical map was projected onto the cortical surface. Shown here are the medial views of the right and left hemispheres of a semi-inflated surface of a template brain. Heatmap color bars range from z-stat=3.1–3.6. The map was cluster corrected for familywise error rate at a whole-brain level with an uncorrected cluster-forming threshold of z=3.1 and corrected extent of *P*<0.05. The full unthresholded map can be viewed here: https://identifiers.org/neurovault.image:869963.

The online version of this article includes the following figure supplement(s) for figure 9:

**Figure supplement 1.** *d-value*, but to a lesser extent *s-value* and the difference between the two, is reflected in BOLD activity in ventromedial prefrontal cortex.

## Representation of revalued values in the ventromedial prefrontal cortex

Several brain areas, in particular the ventromedial prefrontal cortex (vmPFC), have been shown to represent the value of decision alternatives during value-based decisions (*Kennerley et al., 2009*; *Plassmann et al., 2007*; *Bartra et al., 2013*). Based on our finding that the *d-values* provide a better explanation of the behavioral data than the *s-values*, we reasoned that the *d-values* might explain the BOLD activity in these areas beyond that explained by the *s-values*. We included both the *s-value* and the *d-value* of the chosen item in a whole-brain regression analysis of BOLD activity. This parameterization reveals significant correlation of the BOLD signal in the vmPFC with *d-value*, controlling for *s-value* (*Figure 9* and *Supplementary file 1*). In fact, in a separate model that only included *s-value*, the effect of *s-value* on BOLD in the vmPFC did not survive correction for familywise error rate at a whole-brain level (*Figure 9—figure supplement 1b* and *Supplementary file 2* top). In contrast, another model that only included *d-value* revealed a robust effect of *d-value* on BOLD in vmPFC that survived whole-brain correction (*Figure 9—figure supplement 1* and *Supplementary file 2* middle). Finally, to evaluate whether the effect shown in *Figure 9* is not simply captured by the difference in *d-value* and *s-value*, we ran a fourth model that included only (*d-value  s-value*). The effect of this difference between *d-value* and *s-value* on BOLD in vmPFC did not survive whole-brain correction (*Figure 9—figure supplement 1* and *Supplementary file 2* bottom). Collectively, these findings provide additional evidence for revaluation, as capturing a meaningful aspect of the data, in the sense that it accounts for the activity of brain areas known to reflect the value of the choice alternatives.

## Revaluation in other datasets of the food-choice task

To assess the generality of our behavioral results, we applied *Reval* to other publicly available datasets. All involve binary choices between food snacks, similar to *Bakkour et al., 2019*. We analyze data from experiments reported in *Folke et al., 2016* and from the two value-based decision tasks reported in *Sepulveda et al., 2020*.

*Reval* yields results that are largely similar to those observed in the data from *Bakkour et al., 2019*. The values derived from *Reval* led to a better classification of choice difficulty than the explicit value reports (*Figure 10A*). In all three datasets, the $\delta$ values were significantly larger than those obtained from simulated data under the assumption that the values were static and equal to the explicitly reported values (*Figure 10B*). Furthermore, the reduction in the deviance resulting from the application of *Reval* (*Equation 7*) was significantly greater than the reduction observed in simulated data (*Figure 10C*). [All p-values, derived from two-tailed paired t-tests, are shown in the figure].

In the dataset from *Folke et al., 2016*, the deviance was significantly smaller when *Reval* was applied in the forward than in the backward direction, replicating the result in our main experiment. However, in the dataset of *Sepulveda et al., 2020*, no significant difference in deviance was observed (*Figure 10D*). We do not know what explains this discrepancy, although we believe that the differences in experimental design may play a role. In the experiment of *Sepulveda et al., 2020*, unlike the other two datasets that we analyzed, participants performed the experiment in two framing conditions: one in which they chose the item the liked the most, and another one in which the chose the item they disliked the most. These two conditions alternated in short blocks of 40 trials. This alternation may affect valuation in a way that is not captured by the *Reval* algorithm. We expand on this in Discussion.

## Is revaluation a byproduct of deliberation?

We hypothesize that the sequential dependencies we identified with *Reval* may be a corollary of the process by which values are constructed during deliberation. The subjective value of an item depends on the decision-maker's *mindset*, which may change more slowly than the rate of trial presentations. Therefore, the subjective value of an item on a given trial may be informative about the value of the item the next time it is presented. Subjective values are not directly observable, but choices are informative about the items' value.

We assessed the plausibility of this hypothesis with a bounded evidence accumulation model that includes a parameter that controls the correlation between successive evidence samples for a given item. We call this the *correlated-evidence drift-diffusion model* (*ceDDM*). We assume that the decision is resolved by accumulating evidence for and against the different alternatives until a decision threshold is crossed.

The model differs from standard drift-diffusion, where the momentary evidence is a sample drawn from a Normal distribution with expectation equal to $\Delta v_s$ plus unbiased noise, $\mathcal{N}(0, \sqrt{dt})$. Instead, the value of each of the items evolves separately such that the expectations of its value updates are constructed as a Markov chain Monte Carlo (MCMC) process thereby introducing autocorrelation between successive samples of the unbiased noise (see Methods). Crucially, the correlation is not limited to the duration of a trial but extends across trials containing the same item. When an item is repeated in another trial, the process continues to evolve from its value at the time a decision was last made for or against the item.

We fit the model to the data from *Bakkour et al., 2019*. The model was able to capture the relationship between choice, response time and $\Delta v_s$ (*Figure 11A*). *Figure 11B* shows the degree of correlation in the evidence stream as a function of time, for the model that best fit each participant's data. After 1 s of evidence sampling, the correlation was 0.1062 ± 0.0113 (mean ± s.e.m. across participants). This is neither negligible (which would make the model equivalent to the DDM) nor very high (which would render sequential sampling useless, since it can only average out the noise that is not shared across time).

The assumptions embodied by the *ceDDM* are consistent with the results of the *Reval* analysis. We applied the *Reval* algorithm to simulated data obtained from the best-fitting *ceDDM*. The results were in good agreement with the experimental data. The best-fitting $\delta$ values were positive for all participants and in a range similar to what we observed in the data (*Figure 11C*). *Reval* increased the range of RTs when trials were divided by difficulty, implying that *Reval* led to a better classification of easy and difficult decisions (*Figure 11D*). *Reval* applied to the trials in the true order explained the simulated choices better than when applied in the opposite direction (*Figure 11E*). This is because the model assumes that when an item first appears, the last sample obtained for that item was the value reported in the ratings phase for that item. As more samples are obtained for a given item, the correlation with the static values gradually decreases. Additionally, the values obtained from applying *Reval* in the backward direction provided a better explanation of the simulated choices than the static

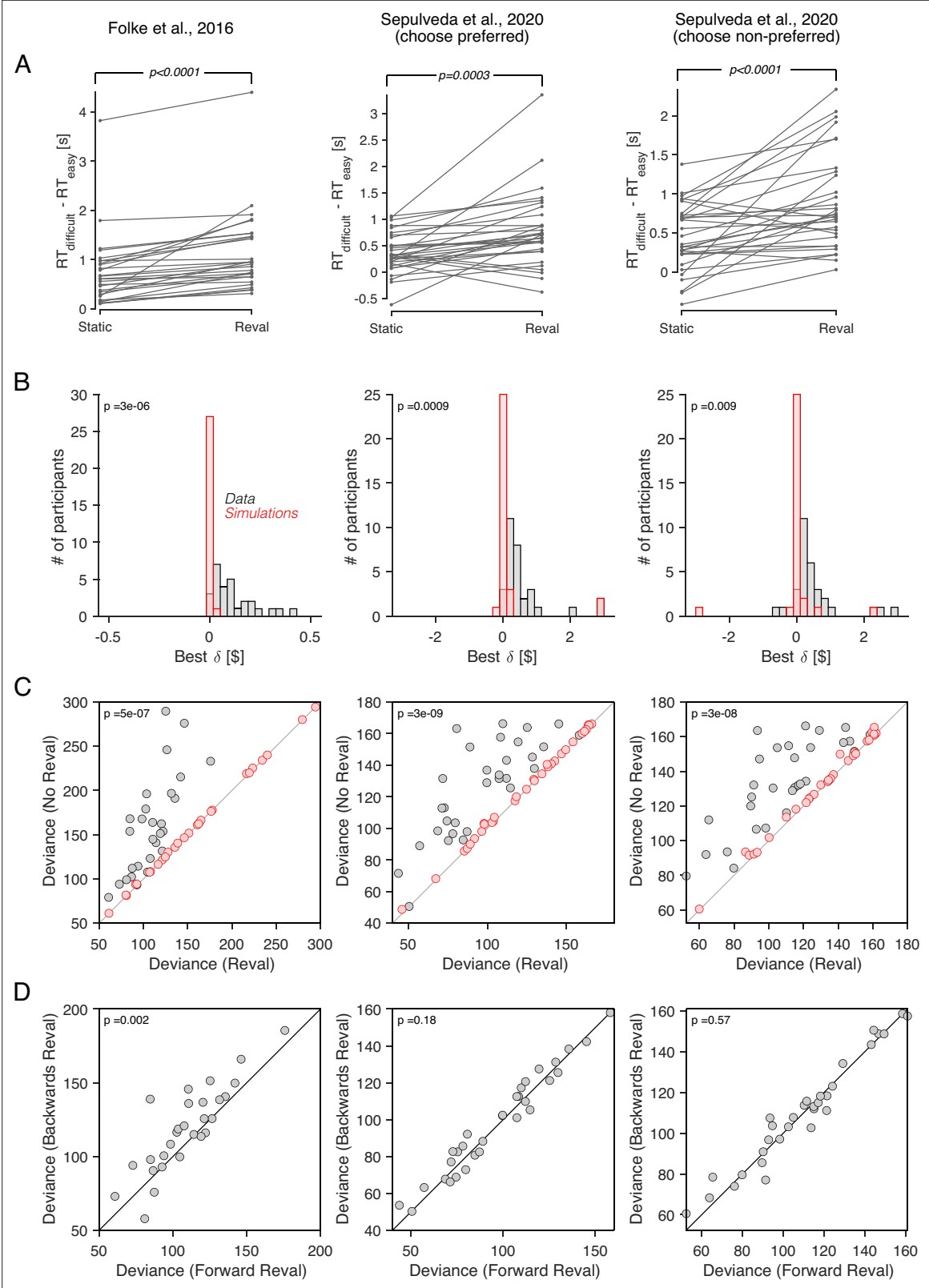

**Figure 10.** Revaluation observed in other datasets. We applied the *Reval* method to other publicly available datasets of the food choice task. In the experiment of ***Folke et al., 2016*** (first column), participants reported their willingness to pay for each of 16 common snack items. In the choice task, they were presented with each unique pair of items and asked to choose the preferred item. Each unique pair was presented twice for a total of 240 trials per participant. In the experiment of ***Sepulveda et al., 2020*** (second and third columns), participants (N=31) reported their willingness to pay for

*Figure 10 continued on next page*

*Figure 10 continued*

each of 60 snack items. They were then presented with pairs of items from which to choose. Pairs were selected based on participants' willingness-to-pay reports to provide comparisons between pairs of high-value, low-value and mixed-value items. The choice task was performed under two framing conditions: *like-framing*, selecting the more preferred item, and *dislike framing*, selecting the less preferred item. The task consisted of six alternating blocks of *like-* and *dislike-framing* (40 trials per block). (**A**) RT difference between *easy* and *difficult* trials, determined as a median split of |Δ$v$|. Same analysis as in *Figure 5C*. (**B**) Histogram of the best-fitting revaluation update ($\delta$) for data simulated by sampling choices from a logistic function fit to the participant's choices (red), and for the actual data (gray). Same analysis as in *Figure 6A*. (**C**) Comparison of the deviance with and without *Reval*. Same analysis as in *Figure 6B*. (**D**) Comparison of the deviance applying *Reval* in the forward and backward directions. Same analysis as in *Figure 7A*. All p-values shown in the figure are from paired t-tests.

values (*Figure 11F*), mirroring the pattern observed observed in the behavioral data (*Figure 7B*). Taken together, the success of *ceDDM* implies that the sequential dependencies we identify with *Reval* may be the result of a value construction process necessary to make a preferential choice.

# Discussion
## Sequential dependencies and choice-induced preference change
We identified sequential dependencies between choices in a value-based decision task. Participants performed a task in which they had to make a sequence of choices among a limited set of items. The best explanation for future choices was obtained by assuming that the subjective value of the chosen

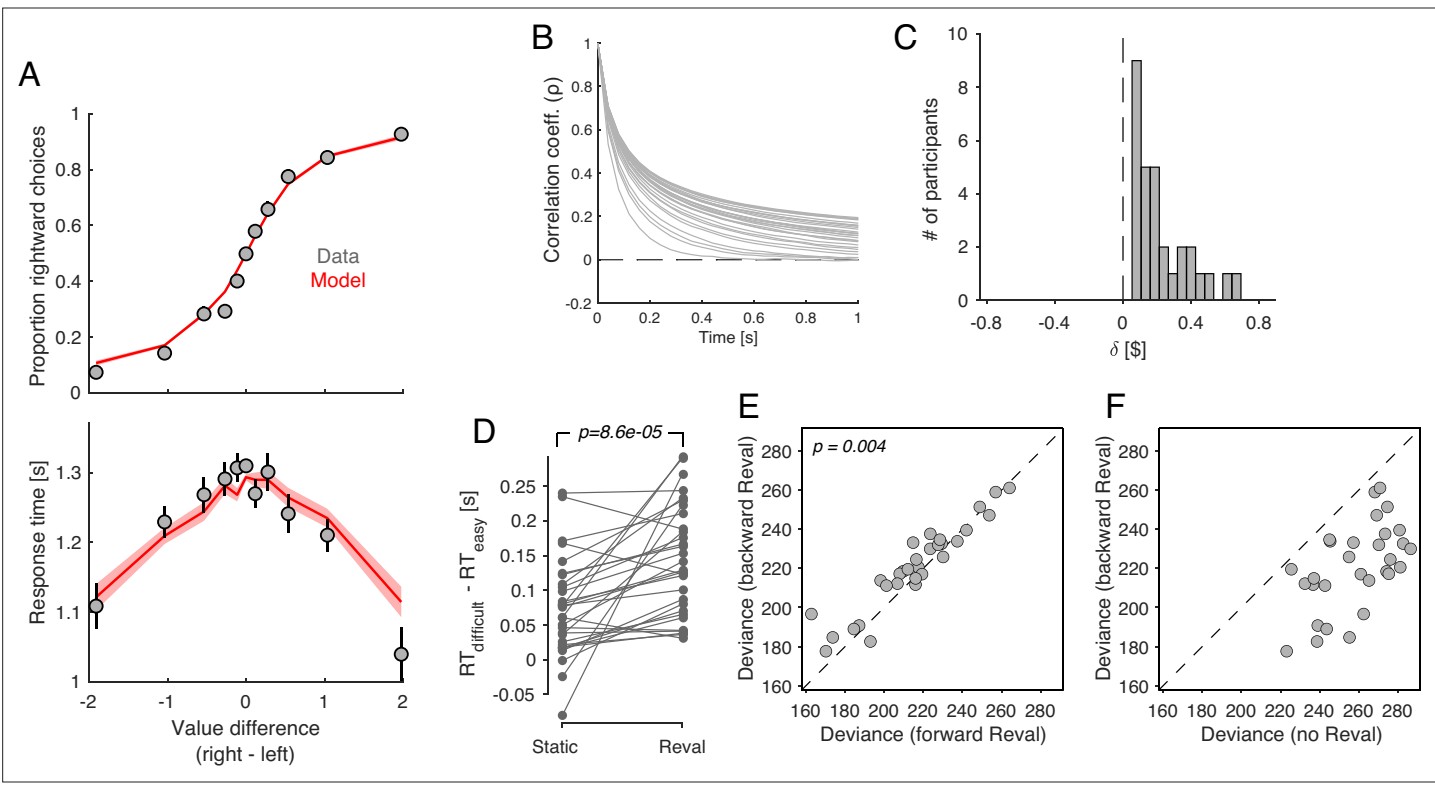

**Figure 11.** Revaluation occurs in a DDM with temporally-correlated noise. A drift-diffusion model with non-independent noise (*ceDDM*) captures the main features of revaluation. (**A**) The *ceDDM* accounts for choices (top) and response times (bottom), plotted as a function of the difference in values obtained from explicit reports ($\Delta v_s$). Same data as in *Figure 1C–D*. Red curves are simulations of the best-fitting model. Each trial was simulated 100 times. Simulations were first averaged within trials and then averaged across trials. Error bars and bands indicate s.e.m. across trials. (**B**) Noise correlations as a function of time lag, obtained from the best-fitting model. Each curve corresponds to a different participant. (**C**) $\delta$ parameters derived by applying *Reval* to simulated data from the best fitting *ceDDM* model to each participant's data. As in the data, $\delta > 0$ for all participants. (**D**) Similar analysis as in *Figure 5C* applied to simulations of the *ceDDM*. As for the data, *Reval* increased the range of RTs obtained after grouping trials by difficulty (by *s-values* on the left and *d-values* on the right; p-value from paired t-test). (**E**) Similar analysis to that of *Figure 7A*, using the simulated data. As observed in the data, the deviance resulting from applying *Reval* in the correct trial order (abscissa) is smaller than when applied in the opposite order (p-value from paired t-test). (**F**) Similar analysis to that of *Figure 7B*, using the simulated data.

item increases and the value of the unchosen item decreases after each decision. Evidence for revaluation was obtained by analyzing the probability that participants make the same decision in pairs of trials with identical options. We also identified revaluation using an algorithm we call *Reval*. The same algorithm allowed us to identify revaluation in other datasets obtained with the food-choice task (*Folke et al., 2016*; *Sepulveda et al., 2020*).

The sequential effects we identified can be interpreted as a manifestation of choice-induced preference change. The usual paradigms for detecting the presence of CIPCs are based on the comparison of value ratings reported before and after a choice (for a review see *Izuma and Murayama, 2013*; *Enisman et al., 2021*). After a difficult decision, the rating of the chosen alternative often increases and that of the rejected alternative often decreases—an effect termed the 'spreading of alternatives'. Many variants of the free choice paradigm have been developed to control for or eliminate the statistical artifact reported by *Chen and Risen, 2010*. One common approach is to compare the 'spreading of alternatives" observed in the free-choice paradigm (rate-choose-rate, or RCR) with a control task in which a different set of participants rate the items twice before the choice phase (RRC). Any spread observed in the RRC condition cannot be explained by the CIPC, since in the RRC condition there is no choice between the two rating phases. The CIPC is measured indirectly, as the difference in the spread of the alternatives between the RCR and the RRC. Other approaches involve asking participants to rate an item that they are led to believe they have chosen, when in fact they have not (*Sharot et al., 2010*; *Johansson et al., 2014*). Any change in ratings cannot be due to the information provided by a choice, since no real choice was made. In addition to the complications introduced by deceiving the participants (e.g. participants may suspect the deception but not mention it to the experimenter), the elimination of a real choice prevents these paradigms from being used to study the process through which subjective values undergo revision during decision formation.

In contrast, our approach to identify changes in value does not require pre- and post-choice ratings. Instead, it requires a sequence of trials in which the same items are presented multiple times (as in *Luettgau et al., 2020*). The revaluation effect we find cannot be explained by the artifact identified by *Chen and Risen, 2010*. Using trials with identical items, we show that the nearer in time the trials with identical items are to each other, the more likely people are to choose the same option. Further, the revaluation algorithm explains choices better when applied in the order in which the trials were presented than when applied in the reverse order. These observations are inconsistent with the notion that item values are fixed (i.e. do not change) during the experiment, regardless of whether values are the same or different from those reported during the rating phase.

## Revaluation during or after deliberation?

We cannot determine with certainty whether the revaluation occurs after the decision or during the deliberation process leading up to the decision. At face value, it might seem that *Reval* implements change after each decision (*Festinger, 1957*). Yet, *Reval* simply identifies a change in value, which may well occur during the deliberation leading to the decision, perhaps owing to a comparison of other items (on other trials) that happen to suggest a dimension of comparison that increases in importance on the current trial (*Lee and Daunizeau, 2020*; *Lichtenstein and Slovic, 2006*). More broadly, the subjective value of an option depends on the *mindset* of the decision maker. This internal state, which in the food-choice task includes aspects such as degree of satiety or sugar craving, can vary over time, causing the value of the items to vary as well. If changes in *mindset* are slow—that is, lasting longer than the duration of a decision—then the value of items will be correlated over time.

We proposed a decision model (*ceDDM*) in which evidence samples are correlated over time. Fitting the model to account for each participant's choices and response times produces a revaluation of magnitude similar to what we observed experimentally. It also predicts that applying *Reval* in the direction in which the trials were presented explains the choices better than applying it in the opposite direction, as we observed in the data. This modeling exercise suggests that the CIPC-like effects we identified may be due to processes that occur during the deliberation leading up to a choice, rather than post-decision processes that attempt to reduce cognitive dissonance. To be clear, we interpret the *ceDDM* only as a proxy for a variety of more nuanced processes. If the *mindset* endures many individual decisions, the subjective value of an item will be correlated over time. While the *ceDDM* captures only a small aspect of this complex process, it has allowed us to explain the sequential dependencies we identified with *Reval*.

The *ceDDM* belongs to a class of sequential sampling models in which the drift rate varies over time. Such models have already been studied in the context of value-based decisions. For example, in the attentional drift-diffusion model (*Krajbich et al., 2010*), the drift rate varies depending on which item is attended, as if the value of the unattended items are discounted by a multiplicative factor. In Dynamic Field Theory (*Busemeyer and Townsend, 1993*), the drift rate varies depending on which attribute is attended. Recently, *Lee and Pezzulo, 2022* showed that a sequential sampling model in which the drift rate varies over time can explain the 'spreading of alternatives' (SoA) characteristic of choice-induced preference change. *Lee and Pezzulo, 2022* propose that the initial rating of the items may be constructed using only the most salient attributes of each item, while in a difficult decision more attributes may be considered, leading to a revaluation that informs the rating reported after the decision phase (see also *Voigt et al., 2019*). Consistent with our proposal, *Lee and Pezzulo, 2022* argue that thinking about non-prominent features during decision-making increases the likelihood that these features will be recalled when evaluating options in subsequent instances.

## More revaluation for the chosen than the unchosen item

We observed that the degree of revaluation was higher for the chosen item than for the unchosen item. This was revealed by a variant of the *Reval* algorithm in which we allowed both items to have different updates. We speculate that this difference can be explained by the asymmetric distribution of attention between the chosen and unchosen items. It is known that the chosen item is looked at longer than the unchosen item (*Krajbich et al., 2010*). Further, CIPC is more likely for items that are remembered to have been chosen or unchosen (*Salti et al., 2014*). So one possibility is that the revaluation is larger for the chosen than for the unchosen item because participants spent more time looking at the chosen item and thus are more likely to remember it, leading to a larger change in value (*Voigt et al., 2019*).

Another possibility derives from the constructive view of preferences and the potential role of attention in decision-making. It is often assumed that value-based decisions involve gathering evidence from different alternatives, and that more evidence is gathered from alternatives that are attended to for longer (*Callaway et al., 2021*; *Li and Ma, 2021*; *Krajbich et al., 2010*). In the *ceDDM*, the correlation in value for a given item decreases with the number of evidence samples collected from the item (*Figure 11B*). Therefore, the more that attention is focused on a given item, the greater the difference between the item's value before and after the decision. Because chosen items are attended to for longer than unchosen items (e.g. *Krajbich et al., 2010*), the chosen item should exhibit larger revaluation than the unchosen one, which is what we observed in the data (*Figure 8*).

## Limitations of our study

One limitation of our study is that we only examined tasks in which static values were elicited from explicit reports of the value of food items. It remains to be determined if other ways of eliciting subjective values (e.g. *Jensen and Miller, 2010*) would lead to similar results. We think so, as the analysis of trials with identical item pairs (*Figure 3*) and the difference between forward and backward *Reval* (*Figure 7A*) are inconsistent with the notion that values are static, regardless of their precise value. It also remains to be determined if our results will generalize to non-food items whose value is less sensitive to satiety and other dynamic bodily states. Perceptual decisions also exhibit sequential dependencies, and it remains to be explored whether these can be explained as a process of value construction, similar to what we propose here for the food-choice task (*Gupta et al., 2024*; *Cho et al., 2002*; *Zylberberg et al., 2018*; *Abrahamyan et al., 2016*).

Another limitation of our study is that, in one of the datasets we analyzed (*Sepulveda et al., 2020*), applying *Reval* in the forward direction was no better than applying it in the backward direction (*Figure 10*). We speculate that this failure is related to idiosyncrasies of the experimental design, in particular, the use of alternating blocks of trials with different instructions (select preferred vs. select non-preferred). More importantly, *Reval* applied in the backward direction led to a significant reduction in deviance relative to that obtained using the static values (*Figure 7B*). This reduction was also observed in the *ceDDM*, suggesting that the effect may be explained by changes in valuation during deliberation. However, we cannot discard a contribution from other, non-dynamic changes in valuation between the rating and choice phase including contextual effects (*Lichtenstein and Slovic,*

*2006*), stochastic variability in explicit value reporting (*Polanía et al., 2019*), and the limited range of numerical scales used to report value.

Finally, we emphasize that the *ceDDM* should be interpreted as a proof-of-principle model used to illustrate how stochastic fluctuations in item desirability can explain many of our results. We chose to model value changes following an MCMC process. However, other stochastic processes or other ways of introducing sequential dependencies (e.g. variability in the starting point of evidence accumulation) may also explain the behavioral observations. Furthermore, there likely are other ways to induce changes in the value of items other than through past decisions. For example, attentional manipulations or other experiences (e.g. actual food consumption) may change one's preference for an item. The current version of the *ceDDM* does not allow for these influences on value, but we see no fundamental limitation to incorporating them in future instantiations of the model.

## Concluding remarks

Our research contributes to a growing body of work exploring the impact of memory on decision-making and preference formation (*Biderman et al., 2020*), and in particular to the CIPC. It has been suggested that the retrieval of an item's value during decision-making renders it susceptible to modification, leading to a revaluation that influences subsequent valuations through a process that has a neural correlate in the hippocampus (*Luettgau et al., 2020*). The link between memorability and preference is also supported by experiments in which the presentation of an item coincides with an unrelated rapid motor response that increases subsequent preference for the item (*Botvinik-Nezer et al., 2021*) and by experiments demonstrating that people prefer items to which they have previously been exposed (*Zajonc, 1968*). As in these studies, ours also highlights the role of memory in revaluation. Due to the associative nature of memory, successive evidence samples are likely to be dependent (*Rhodes and Turvey, 2007*). A compelling illustration of this effect was provided by Elias Costa and colleagues (*Costa et al., 2009*). Participants were asked to report the first word that came to mind when presented with a word generated by another participant, which was then shown to yet another participant. The resulting chain resembled Lévy flights in semantic space, characterized by mostly short transitions to nearby words and occasional large jumps. Similar dynamic processes have been used to describe eye movements during visual search (*Bella-Fernández et al., 2022*) and the movement of animals during reward foraging (*Brown et al., 2007*; *Hills et al., 2015*). It is intriguing to consider that a similar process may describe how decision-makers search their memory for evidence that bears on a decision.

## Methods

### Food choice task

A total of 30 participants completed the snack task, which consisted of a rating and a choice phase. The experimental procedures were approved by the Institutional Review Board (IRB) at Columbia University, and participants provided signed informed consent before participating in the study. The data were previously published in *Bakkour et al., 2019*.

### Rating phase

Participants were shown a series of snack items in a randomized order on a computer screen. They indicated their willingness to pay (WTP) by using the computer mouse to move a cursor along an analog scale ranging from $0 to $3 at the bottom of the screen. The process was self-paced, and each snack item was presented one at a time. After completing the ratings for all 60 items, participants were given the opportunity to revise their ratings. The 60 items were re-displayed in random order, with the original bids displayed below each item. Participants either chose to keep their original bid by clicking 'NO' or to revise the bid by clicking 'YES', which re-displayed the analog scale for bid adjustment. We take the final WTP that is reported for each item as the corresponding *static* value (*s-value*).

### Choice phase

From the 60 rated items, 150 unique pairs were formed, ensuring variation in $\Delta v_s$. Each of the 60 items was included in five different pairs. Sixty item pairs were presented twice, resulting in a total

of 210 trials per participant. Item pairs were presented in random order, with one item on each side of a central fixation cross. Participants were instructed to select their preferred food item and were informed that they would receive their chosen food from a randomly selected trial to consume at the end of the experiment. The task took place in an MRI scanner. Participants indicated their choice on each trial by pressing one of two buttons on an MRI-compatible button box. They had up to 3 s to make their choice. Once a choice was made, the chosen item was highlighted for 500ms. Trials were separated by an inter-trial interval (ITI) drawn from a truncated exponential distribution with a minimum ITI of 1 and a maximum ITI of 12 s. The resulting distribution of ITIs across trials had a true mean of 3.05 s and a standard deviation of 2.0 s.

## Data analysis

Association between the *s-values*, choice and RT. We used the following logistic regression model to evaluate the association between the *s-values* and the probability of choosing the item on the right:

$$\text{logit}[p_{\text{right}}] = \sum_{i=1}^{N_{\text{subj}}} \beta_{0,i} I_i + \beta_1 \Delta v_s, \tag{2}$$

where $I_i$ is an indicator variable that takes the value 1 if the trial was completed by subject $i$ and 0 otherwise. We used a t-test to evaluate the hypothesis that the corresponding regression coefficient is zero, using the standard error of the estimated regression coefficient.

Similarly, we used a linear regression model to test the influence of $\Delta v_s$ on response times:

$$\text{RT} = \sum_{i=1}^{N_{\text{subj}}} \beta_{0,i} I_i + \beta_1 \left| \Delta v_s \right| + \beta_2 \Sigma v_s, \tag{3}$$

where $| \cdot |$ denotes absolute value and $\Sigma v_s$ is the sum of the value of the two items presented on each trial. The last term was included to account for the potential influence of value sum on response time (**Smith and Krajbich, 2019**).

### Predicting choices in *cynosure* trials

We used two logistic regression models to predict the choice in each trial using observations from the other trials. We refer to the trial under consideration as the *cynosure* trial (**Figure 2**). One model uses the explicitly reported values:

$$\text{logit}[p_{\text{right}}] = \beta_0 + \beta_1 \Delta v_s, \tag{4}$$

while the other model uses the choices made on other trials:

$$\text{logit}[p_{\text{right}}] = \beta_0 + \sum_{i=1}^{N_{\text{items}}} \beta_i f(i), \tag{5}$$

where

$$f(i) = \begin{cases} 1 & \text{if item } i \text{ is on the right} \\ -1 & \text{if item } i \text{ is on the left} \\ 0 & \text{otherwise} \end{cases} \tag{6}$$

For this model, we included an L2 regularization with $\lambda = 0.5$. Both models were fit independently for each participant. We only included trials with the first appearance of each item pair (i.e. we did not include the repeated trials) so that the choice prediction for the *cynosure* trial is not influenced by the choice made in the paired trial containing the same items as in the *cynosure* trial.

Association between *d-values* and choice. We tested the association between *d-values* and choice with a logistic regression model fit to the choices. We included separate regressors for $\Delta v_d$ and $\Delta v_s$:

$$\text{logit}[p_{\text{right}}] = \beta_0 + \beta_s \Delta v_s + \beta_d \Delta v_d \tag{7}$$

The model was fit separately for each participant. *Figure 5—figure supplement 1* shows the regression coefficients associated with $\Delta v_s$ and $\Delta v_d$.

## Choice and response time functions

When plotting the psychometric and chronometric functions (e.g., *Figure 1C–D*), we binned trials depending on the value of $\Delta v_s$ (or $\Delta v_d$). The bins are defined by the following edges: { $-\infty, -1.5, -0.75, -0.375, -0.1875, -0.0625, 0.0625, 0.1875, 0.375, 0.75, 1.5, \infty$ }. We averaged the choice or RT for the trials (grouped across participants) within each bin and plotted them aligned to the mean $\Delta v_x$ of each bin.

## Match probability

We used logistic regression to determine if the probability of giving the same response to the pair of trials with identical stimuli depended on the number of trials in between (*Figure 3*). The model is:

$$\text{logit}[p_{match}] = \sum_{i=1}^{N_{subj}} \beta_{0,i} I_i + \sum_{i=1}^{N_{subj}} \beta_{1,i} I_i |\Delta v_s| + \beta_2 \left( T_{2nd} - T_{1st} \right) \tag{8}$$

where $p_{match}$ is the probability of choosing the same item on both occasions, $I_i$ is an indicator variable that takes a value of 1 if the pair of trials correspond to subject $i$, and zero otherwise, and $T_{1st}$ and $T_{2nd}$ are the trial number of the first and second occurrences of the same pair, respectively. We used a t-test to evaluate the hypothesis that $\beta_2 = 0$ (i.e., that the separation between trials with identical stimuli had no effect on $p_{match}$).

## Drift-diffusion model

We fit the choice and RT data with a drift-diffusion model. It assumes that the decision variable, $x$, is given by the accumulation of signal and noise, where the signal is a function of the difference in value between the items, $\Delta v$, and the noise is equal to $\sqrt{dt}$, where $dt$ is the time step, such that the accumulated noise after 1 s of unbounded accumulation, the variance of the accumulated noise is equal to 1. The decision variable follows the difference equation,

$$x_{t+1} = x_t + \kappa \, dt \, (\mu + \mu_0) + \sqrt{dt} \, \eta_t, \tag{9}$$

where $\eta_t$ is sampled from a normal distribution with a mean 0 and variance 1, $\kappa$ is a signal-noise parameter, $\mu$ is the drift rate and $\mu_0$ is a bias coefficient that is included to account for potential asymmetries between right and left choices.

We assume that the drift rate is a (potentially nonlinear) function of $\Delta v$. We parameterize this relationship as a power law, so that

$$\mu = \text{sign}(\Delta v) |\Delta v|^{\gamma}, \tag{10}$$

where sign is the sign operation, || indicates absolute value, and $\gamma$ is a fit parameter.

The decision terminates when the accumulated evidence reaches an upper bound, signaling a rightward choice, or a lower bound, signaling a leftward choice. The bound is assumed to collapse over time. It is constant until time $d$, and then it collapses at rate $a$:

$$B(t) = \pm \begin{cases} B_0 & \text{if } t < d \\ B_0 \exp^{-a(t-d)} & \text{otherwise.} \end{cases} \tag{11}$$

Collapsing bounds are needed to explain why choices that are consistent with the value ratings are usually faster than inconsistent choices for the same $\Delta v$.

The response time is the sum of the the decision time, given by the time taken by the diffusing particle to reach of the bounds, and a non-decision time which is assumed to be normally distributed with mean $\mu_{nd}$ and standard deviation $\sigma_{nd}$.

The model has 8 parameters: $\{\kappa, B_0, a, d, \gamma, \mu_0, \mu_{nd}, \sigma_{nd}\}$. The standard deviation of the non-decision times ($\sigma_{nd}$) was fixed to 0.05 s. For the fits shown in **Figures 1C–D , and 5A**, we fit the model to grouped data from all participants. For the analysis of variance explained (**Figure 5**) and model comparison (**Figure 5—figure supplement 2**), we fit the model separately for each participant. The model was fit to maximize the log of the likelihood of the parameters given the single-trial choice and RT:

$$\log L(\text{parameters}) = \sum_{i=1}^{n_{\text{trials}}} \log \left( p \left( \text{choice}^{(i)}, \text{RT}^{(i)} | \Delta v^{(i)}, \text{parameters} \right) \right). \quad (12)$$

We evaluate the likelihood by numerically solving the Fokker-Planck (FP) equation that described the dynamics of the drift-diffusion process, using the Chang-Cooper fully-implicit method (**Chang and Cooper, 1970**; **Kiani and Shadlen, 2009**; **Zylberberg et al., 2016**). For computational considerations, we bin the values of $\Delta v$ to multiples of $0.1. From the numerical solution of the FP equation, we obtain the distribution of decision times, which is convolved with the truncated Gaussian distribution of non-decision latencies. The truncation ensures that the non-decision times are non-negative, which could otherwise occur during the optimization process for large values of $\sigma_{nd}$. The parameter search was performed using the Bayesian Adaptive Direct Search (BADS) algorithm (**Acerbi and Ma, 2017**).

## Revaluation algorithm

The *Reval* algorithm was applied to each participant independently. The values are initialized to those reported during the ratings phase. They are then revised, based on the outcome of each trial, in the order of the experiment.

The value of the chosen item is increased by $\delta$ and the value of the unchosen item is decreased by the same amount. The revaluation affects future decisions in which the same item is presented.

We searched for the value of $\delta^*$ that minimizes the deviance of the logistic regression model specified by **Equation 1**. The model's deviance is given by:

$$\text{DEV} = \sum_{i=1}^{N_{\text{tr}}} 2 \log_e \left( \frac{1}{\hat{c}_i} \right) \quad (13)$$

where the sum is over trials and $\hat{c}_i$ is the probability assigned to the choice on trial $i$ obtained from the best-fitting logistic regression model.

We complemented this iterative algorithm with a second approach that estimates $\delta^*$ using the history of choices preceding each trial. Nearly identical $\delta$ values are derived using a single logistic regression model in which the binary choice made on each trial depends on the number of times each of the two items was selected and rejected on previous trials. The model is:

$$\text{logit}[p_{\text{right}}] = \sum_{i=1}^{N_{\text{subj}}} \beta_{0,i} I_i + \sum_{i=1}^{N_{\text{subj}}} \beta_{1,i} I_i \Delta v_s + \sum_{i=1}^{N_{\text{subj}}} \beta_{2,i} I_i \Delta_{ch} \quad (14)$$

where, as before, $I_i$ is an indicator variable that takes a value of 1 if the trial was completed by subject $i$ and 0 otherwise. The key variable is $\Delta_{ch}$. It depends on the number of past trials in which the item presented on the right in the current trial was chosen ($n_{\text{ch}}^{\text{right}}$) and not chosen ($n_{\neg\text{ch}}^{\text{right}}$), and similarly, the number of past trials in which the item presented on the left in the current trial was chosen ($n_{\text{ch}}^{\text{left}}$) and not chosen ($n_{\neg\text{ch}}^{\text{left}}$):

$$\Delta_{ch} = n_{\text{ch}}^{\text{right}} - n_{\neg\text{ch}}^{\text{right}} + n_{\neg\text{ch}}^{\text{left}} - n_{\text{ch}}^{\text{left}}. \quad (15)$$

The variable $\Delta_{ch}$ represents the influence of past choices. The signs in **Equation 15** are such that a positive (negative) value of $\Delta_{ch}$ indicates a bias toward the right (left) item. To obtain the $\delta^*$ in units equivalent to those derived with *Reval*, we need to divide the regression coefficient $\beta_{2,i}$ by the sensitivity coefficient $\beta_{1,i}$, separately for each subject $i$. As can be seen in **Figure 5—figure supplement 3**, the values obtained with this method are almost identical to those obtained with the *Reval* algorithm.

## Correlated-evidence DDM

The model assumes that at each moment during the decision-making process, the decision-maker can only access a noisy sample of the value of each item. These samples are normally distributed, with parameters such that their unbounded accumulation over one second is also normally distributed with a mean equal to $\kappa v_s$, where $v_s$ is the explicit value reported during the Ratings phase and $\kappa$ is a measure of signal-to-noise, and a standard deviation equal to 1.

Crucially, for each item, the noise in successive samples is correlated. To generate the correlated samples, we sample from a Markov chain using the Metropolis-Hastings algorithm (**Chib and Greenberg, 1995**). The target distribution is the normally distributed value function described in the previous paragraph. The proposal density is also normally distributed. Its width determines the degree of correlation between consecutive samples. Typically, the correlation between successive samples is considered a limitation of the Metropolis-Hastings algorithm. Here, however, it allows us to generate correlated samples from a target distribution. The standard deviation of the proposal density is $\sqrt{dt}/\tau$. Higher values of $\tau$ result in a narrower proposal density, hence more strongly correlated samples. We sample from the same Markov chain across different trials in which the same item is presented, so that the last sample obtained about an item in a given trial is the initial state of the Markov chain the next time the item is presented.

At each moment ($dt = 40ms$), we sample one value for the left item and another for the right item, compute their difference (right minus left), and accumulate this difference until it crosses a threshold at $+B_0$, signaling a rightward choice, or at $-B_0$, signaling a leftward choice. The decision time is added to the non-decision time, $\mu_{nd}$, to obtain the response time.

We fit the model to the data as follows. For each item, we simulate many Markov chains. In each trial, $i$, we take samples from each chain until the accumulation of these samples reaches one of the two decision thresholds. Then we calculate the likelihood ($L$) of obtaining the choice and the RT displayed by the participant on that trial as:

$$L(\text{choice}_i, \text{RT}_i) = \frac{1}{N} \sum_{j=1}^{N} L_j(\text{choice}_i, \text{RT}_i)$$

$$L_j(\text{choice}_i, \text{RT}_i) = \mathbb{1}_{i,j} \mathcal{N}(\text{RT}_i | \text{RT}_i^{(j)}, \sigma_{nd})$$

(16)

where $N = 1,000$ is the number of Markov chains, $\mathbb{1}_{i,j}$ is an indicator function that takes the value 1 if the choice made on chain $j$ is the same as the choice made by the participant on trial $i$ and 0 otherwise, $\mathcal{N}(x|y,z)$ is the normal probability density function with mean $y$ and standard deviation $z$ evaluated at $x$, and $\sigma_{nd}$ is a parameter fit to the data.

When an item is presented again in a future trial, the initial state of each Markov chain depends on the state it was in the last time the item was presented. The initial state of each chain is obtained by sampling 1000 values (one per chain) from the distribution given by the final state of each chain. The sampling is weighted by the value of $L_j$ of each chain (**Equation 16**), so that chains that better explained the choice and RT in the last trial are more likely to be sampled from in future trials.

The model has 5 parameters per participant: $\{\kappa, B_0, \tau, \mu_{nd}, \sigma_{nd}\}$, which were fit to maximize the sum, across trials, of the log of $L$ using BADS (**Acerbi and Ma, 2017**).

The correlations in **Figure 11B** were generated using the best-fitting parameters for each participant to simulate 100,000 Markov chains. We generate Markov chain samples independently for the left and right items over a 1 s period. To illustrate noise correlations, the simulations assume that the static value of both the left and right items is zero. We then calculate the difference in dynamic value ($x$) between the left and right items at each time ($t$) and for each of the Markov chains ($i$). Pearson's correlation is computed between these differences at time zero, $x_i(t=0)$, and at time $x_i(t=\tau)$, for different time lags $\tau$. Correlations were calculated independently for each participant. Each trace in **Figure 11B** represents a different participant.

## fMRI analysis

### Acquisition

Imaging data were acquired on a 3T GE MR750 MRI scanner with a 32-channel head coil. Functional data were acquired using a T2*-weighted echo planar imaging sequence (repetition time (TR)=2 s,

echo time (TE)=22ms, flip angle (FA) = 70°, field of view (FOV)=192 mm, acquisition matrix of 96x96). Forty oblique axial slices were acquired with a 2 mm in-plane resolution positioned along the anterior commissure-posterior commissure line and spaced 3 mm to achieve full brain coverage. Slices were acquired in an interleaved fashion. We acquired three runs of the food choice task, each composed of 70 trials. Each of the food choice task functional runs consisted of 212 volumes and lasted 7 min. In addition to functional data, a single three-dimensional high-resolution (1 mm isotropic) T1-weighted full-brain image was acquired using a BRAVO pulse sequence for brain masking and image registration.

## Preprocessing

Raw DICOM files were converted into Nifti file format and organized in the Brain Imaging Data Structure (BIDS) using dcm2niix (*Li et al., 2016*). Results included in this manuscript come from preprocessing performed using *fMRIPrep* 22.1.1 (*Esteban et al., 2019*; *Esteban et al., 2018*; RRID:SCR_016216), which is based on *Nipype* 1.8.5 (*Gorgolewski et al., 2011*; *Gorgolewski et al., 2018*; RRID:SCR_002502).

## Anatomical data preprocessing

The T1-weighted (T1w) image was corrected for intensity non-uniformity (INU) with `N4BiasFieldCorrection` (*Tustison et al., 2010*), distributed with ANTs 2.3.3 (*Avants et al., 2008*, RRID:SCR_004757), and used as T1w-reference throughout the workflow. The T1w-reference was then skull-stripped with a *Nipype* implementation of the `antsBrainExtraction.sh` workflow (from ANTs), using OASIS-S30ANTs as target template. Volume-based spatial normalization to one standard space (MNI152N-Lin2009cAsym) was performed through nonlinear registration with `antsRegistration` (ANTs 2.3.3), using brain-extracted versions of both T1w reference and the T1w template. The following template was selected for spatial normalization: *ICBM 152 Nonlinear Asymmetrical template version 2009c* [*Fonov et al., 2009*, RRID:SCR_008796; TemplateFlow ID: MNI152NLin2009cAsym].

## Functional data preprocessing

For each of the three BOLD runs per subject, the following preprocessing was performed. First, a reference volume and its skull-stripped version were generated using a custom methodology of *fMRIPrep*. Head-motion parameters with respect to the BOLD reference (transformation matrices, and six corresponding rotation and translation parameters) are estimated before any spatiotemporal filtering using `mcflirt` (FSL 6.0.5.1:57b01774, *Jenkinson et al., 2002*). The BOLD time-series (including slice-timing correction when applied) were resampled onto their original, native space by applying the transforms to correct for head-motion. These resampled BOLD time-series will be referred to as *preprocessed BOLD in original space*, or just *preprocessed BOLD*. The BOLD reference was then co-registered to the T1w reference using `mri_coreg` (FreeSurfer) followed by `flirt` (FSL 6.0.5.1:57b01774, *Jenkinson and Smith, 2001*) with the boundary-based registration (*Greve and Fischl, 2009*) cost-function. Co-registration was configured with six degrees of freedom. Several confounding time-series were calculated based on the *preprocessed BOLD*: framewise displacement (FD) and DVARS. FD was computed using two formulations following Power (absolute sum of relative motions, *Power et al., 2014*) and Jenkinson (relative root mean square displacement between affines, *Jenkinson et al., 2002*). FD and DVARS are calculated for each functional run, both using their implementations in *Nipype* (following the definitions by *Power et al., 2014*). The head-motion estimates calculated in the correction step were also placed within the corresponding confounds file. The confound time series were derived from head motion estimates (*Satterthwaite et al., 2013*). Frames that exceeded a threshold of 0.5 mm FD or 1.5 standardized DVARS were annotated as motion outliers. The BOLD time-series were resampled into standard space, generating a *preprocessed BOLD run in MNI152N-Lin2009cAsym space*. First, a reference volume and its skull-stripped version were generated using a custom methodology of *fMRIPrep*. All resamplings can be performed with *a single interpolation step* by composing all the pertinent transformations (i.e. head-motion transform matrices, susceptibility distortion correction when available, and co-registrations to anatomical and output spaces). Gridded (volumetric) resamplings were performed using antsApplyTransforms (ANTs), configured with Lanczos interpolation to minimize the smoothing effects of other kernels (*Lanczos, 1964*).

Many internal operations of *fMRIPrep* use *Nilearn* 0.9.1 (*Abraham et al., 2014*, RRID:SCR_001362), mostly within the functional processing workflow. For more details of the pipeline, see the section corresponding to workflows in *fMRIPrep*'s documentation.

## Analysis

We conducted a GLM analysis to look at BOLD activity related to *d-values*, *s-values*, and the difference between the two. We ran four separate models.

*Main fMRI Model* included five regressors: (*i*) onsets for all valid trials, modeled with a duration equal to the average RT across all valid choices and participants; (*ii*) same onsets and duration as (*i*) modulated by RT demeaned across these trials within each run for each participant; (*iii*) same onsets and duration as (*i*) but modulated by the *s-value* of the chosen item demeaned across trials within each run for each participant; (*iv*) same onsets and duration as (*i*) but modulated by the *d-value* of the chosen item demeaned across these trials within each run for each participant; (*v*) onsets for missed trials. The map in *Figure 9* was generated using this model.

fMRI Model of s-value only included four regressors; all but regressor (*iv*) in *Main fMRI Model*. The map in *Figure 9—figure supplement 1* top was generated using this model.

fMRI model of d-value only included four regressors; all but regressor (*iii*) in *Main fMRI Model*. The map in *Figure 9—figure supplement 1* middle was generated using this model.

fMRI model of d-value s-value only included four regressors; regressors (*i*) and (*ii*) were the same as in *Main fMRI Model*, regressor (*iii*) had the same onsets and duration as (*i*) but modulated by (*d-value s-value*) of the chosen item demeaned across trials within each run for each participant, and regressor (*iv*) included onsets for missed trials. The map in *Figure 9—figure supplement 1* bottom was generated using this model.

All four models included the six x, y, z translation and rotation motion parameters, FD, DVARS, and motion outliers obtained from textitfmriprep (described above) as confound regressors of no interest. All regressors were entered at the first level of analysis, and all (except the added confound regressors) were convolved with a canonical double-gamma hemodynamic response function. The time derivative of each regressor (except the added confounding regressors) was included in the model. No orthogonalization between regressors was performed. Models were estimated separately for each participant and run.

GLMs were estimated using FSL's FMRI Expert Analysis Tool (FEAT). The first-level time-series GLM analysis was performed for each run per participant using FSL's FILM. The first-level contrast images were then combined across runs per participant using fixed effects. The group-level analysis was performed using FMRIB's Local Analysis of Mixed Effects (FLAME1; *Beckmann et al., 2003*). Group-level maps were corrected to control the family-wise error rate using cluster-based Gaussian random field correction for multiple comparisons, with an uncorrected cluster-forming threshold of z=3.1 and corrected extent threshold of p<0.05.

## Acknowledgements

We thank Ari Pakman for helpful discussions.

This work was supported by the National Institutes of Health (R01NS113113 to MNS and MH121093 to DS), the Air Force Office of Scientific Research under award (FA9550-22-1-0337 to MNS), the Howard Hughes Medical Institute (MNS), The McKnight Foundation Memory and Cognitive Disorders Award (DS), and the National Science Foundation (1822619 to DS and 1606916 to AB).

## Additional information

### Funding

| Funder | Grant reference number | Author |
| --- | --- | --- |
| National Institutes of Health | R01NS113113 | Michael N Shadlen |

| Funder | Grant reference number | Author |
|---|---|---|
| Air Force Office of Scientific Research | FA9550-22-1-0337 | Michael N Shadlen |
| Howard Hughes Medical Institute | | Michael N Shadlen |
| McKnight Foundation | | Daphna Shohamy |
| National Science Foundation | 1606916 | Akram Bakkour |
| National Science Foundation | 1822619 | Daphna Shohamy |
| National Institutes of Health | MH121093 | Daphna Shohamy |

The funders had no role in study design, data collection and interpretation, or the decision to submit the work for publication.

### Author contributions

Ariel Zylberberg, Conceptualization, Data curation, Software, Formal analysis, Validation, Investigation, Visualization, Methodology, Writing – original draft, Writing – review and editing; Akram Bakkour, Data curation, Software, Formal analysis, Validation, Investigation, Visualization, Methodology, Writing – review and editing; Daphna Shohamy, Michael N Shadlen, Resources, Supervision, Funding acquisition, Project administration, Writing – review and editing

### Author ORCIDs

Ariel Zylberberg (b) https://orcid.org/0000-0002-2572-4748
Michael N Shadlen (b) https://orcid.org/0000-0002-2002-2210

Reviewer #1 (Public review): https://doi.org/10.7554/eLife.96997.3.sa1
Reviewer #2 (Public review): https://doi.org/10.7554/eLife.96997.3.sa2
Author response https://doi.org/10.7554/eLife.96997.3.sa3

## Additional files

### Supplementary files

• MDAR checklist

• Supplementary file 1. Activation table for map in *Figure 9*. The effect of d-value on BOLD in Main fMRI model. For each cluster, the list shows regions from the Harvard-Oxford atlas that contained a peak activation of a subcluster, along with the peak p-value, the peak effect size, and the peak X/Y/Z location for the cluster in MNI space.

• Supplementary file 2. Activation tables for maps in *Figure 9—figure supplement 1*. The effect of s-value on BOLD in fMRI Model of s-value only (top), the effect of d-value on BOLD in fMRI Model of d-value only (middle), and the effect of (s-value − d-value) in fMRI model of (d-value − s-value) only (bottom). For each cluster, the list shows regions from the Harvard-Oxford atlas that contained a peak activation of a subcluster, along with the peak p-value, the peak effect size, and the peak X/Y/Z location for the cluster in MNI space.

### Data availability

The data and code required to reproduce the analyses and figures are available on GitHub, (copy archived at *Zylberberg, 2024*).

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
