## [Editor Report · eLife Assessment]

This **important** study addresses key assumptions underlying current models of the formation of value-based decisions. The authors provide **convincing** evidence that the subjective values human participants assign to items change across sequences of multiple decisions. They establish methods to detect these changes in frequently used behavioral task designs.

---

## [Referee Report · Reviewer #1 (Public review)]

Summary:

There is a long-standing idea that choices influence evaluation: options we choose are re-evaluated to be better than they were before the choice. There has been some debate about this finding, and the authors developed several novel methods for detecting these re-evaluations in task designs where options are repeatedly presented against several alternatives. Using these novel methods the authors clearly demonstrate this re-evaluation phenomenon in several existing datasets and show that estimations of dynamic valuation correlate with neural activity in prefrontal cortex.

Strengths:

The paper is well-written and figures are clear. The authors provided evidence for the behaviour effect using several techniques and generated surrogate data (where the ground truth is known) to demonstrate the robustness of their methods. The author avoid over-selling the work, with a lucid description of limitations, and potential for further exploration of the work, in the discussion.

Comments on revisions:

The authors did a good job responding to the comments.

---

## [Referee Report · Reviewer #2 (Public review)]

Zylberberg and colleagues show that food choice outcomes and BOLD signal in the vmPFC are better explained by algorithms that update subjective values during the sequence of choices compared to algorithms based on static values acquired before the decision phase. This study presents a valuable means of reducing the apparent stochasticity of choices in common laboratory experiment designs. The evidence supporting the claims of the authors is solid, although currently limited to choices between food items because no other goods were examined. The work will be of interest to researchers examining decision making across various social and biological sciences.

Comments on revisions:

We thank the authors for carefully addressing our concerns about the first version of the manuscript. The manuscript text and contributions are now much more clear and convincing.

---

## [Author Response]

The following is the authors’ response to the original reviews.

**Public Reviews:**

**Reviewer #1 (Public Review):**
Summary:There is a long-standing idea that choices influence evaluation: options we choose are re-evaluated to be better than they were before the choice. There has been some debate about this finding, and the authors developed several novel methods for detecting these re-evaluations in task designs where options are repeatedly presented against several alternatives. Using these novel methods the authors clearly demonstrate this re-evaluation phenomenon in several existing datasets.Strengths:The paper is well-written and the figures are clear. The authors provided evidence for the behaviour effect using several techniques and generated surrogate data (where the ground truth is known) to demonstrate the robustness of their methods.Weaknesses:The description of the results of the fMRI analysis in the text is not complete: weakening the claim that their re-evaluation algorithm better reveals neural valuation processes.

We appreciate the reviewer’s comment regarding the incomplete account of the fMRI results. In response, we implemented Reviewer #2's suggestion to run additional GLM models for a clearer interpretation of our findings. We also took this opportunity to apply updated preprocessing to the fMRI data and revise the GLM models, making them both simpler and more comprehensive. The results section is thus substantially revised, now including a new main figure and several supplemental figures that more clearly present our fMRI findings. Additionally, we have uploaded the statistical maps to NeuroVault, allowing readers to explore the full maps interactively rather than relying solely on the static images in the paper. The new analyses strengthen our original conclusion: dynamic values (previously referred to as revalued values, following the reviewer’s suggestion) better explain BOLD activity in the ventromedial prefrontal cortex, a region consistently associated with valuation, than static values (values reported prior to the choice phase in the auction procedure).

**Reviewer #2 (Public Review):**
Summary:Zylberberg and colleagues show that food choice outcomes and BOLD signal in the vmPFC are better explained by algorithms that update subjective values during the sequence of choices compared to algorithms based on static values acquired before the decision phase. This study presents a valuable means of reducing the apparent stochasticity of choices in common laboratory experiment designs. The evidence supporting the claims of the authors is solid, although currently limited to choices between food items because no other goods were examined. The work will be of interest to researchers examining decision-making across various social and biological sciences.Strengths:The paper analyses multiple food choice datasets to check the robustness of its findings in that domain.The paper presents simulations and robustness checks to back up its core claims.Weaknesses:To avoid potential misunderstandings of their work, I think it would be useful for the authors to clarify their statements and implications regarding the utility of item ratings/bids (e-values) in explaining choice behavior. Currently, the paper emphasizes that e-values have limited power to predict choices without explicitly stating the likely reason for this limitation given its own results or pointing out that this limitation is not unique to e-values and would apply to choice outcomes or any other preference elicitation measure too. The core of the paper rests on the argument that the subjective values of the food items are not stored as a relatively constant value, but instead are constructed at the time of choice based on the individual's current state. That is, a food's subjective value is a dynamic creation, and any measure of subjective value will become less accurate with time or new inputs (see Figure 3 regarding choice outcomes, for example). The e-values will change with time, choice deliberation, or other experiences to reflect the change in subjective value. Indeed, most previous studies of choice-induced preference change, including those cited in this manuscript, use multiple elicitations of e-values to detect these changes. It is important to clearly state that this paper provides no data on whether e-values are more or less limited than any other measure of eliciting subjective value. Rather, the paper shows that a static estimate of a food's subjective value at a single point in time has limited power to predict future choices. Thus, a more accurate label for the e-values would be static values because stationarity is the key assumption rather than the means by which the values are elicited or inferred.

Thank you for this helpful comment. We changed the terminology following the reviewer’s suggestion. The “explicit” values (e-values or ve) are now called “static” values (s-values or vs). Accordingly, we also changed the “Reval” values (r-values or vr) to “dynamic” values (d-values or vd).

We also address the reviewer's more general point about the utility of item ratings/bids (s-values) and whether our results are likely to hold with other ways of eliciting subjective values. We added a new sub-section in Discussion addressing this and other limitations of our study. To address the reviewer’s point, we write:

“One limitation of our study is that we only examined tasks in which static values were elicited from explicit reports of the value of food items. It remains to be determined if other ways of eliciting subjective values (e.g., Jensen and Miller, 2010) would lead to similar results. We think so, as the analysis of trials with identical item pairs (Fig. 3) and the difference between forward and backward *Reval* (Fig. 7) are inconsistent with the notion that values are static, regardless of their precise value. It also remains to be determined if our results will generalize to non-food items whose value is less sensitive to satiety and other dynamic bodily states. Perceptual decisions also exhibit sequential dependencies, and it remains to be explored whether these can be explained as a process of value construction, similar to what we propose here for the food-choice task (Gupta et al., 2024; Cho et al., 2002; Zylberberg et al., 2018; Abrahamyan et al., 2016).”

There is a puzzling discrepancy between the fits of a DDM using e-values in Figure 1 versus Figure 5. In Figure 1, the DDM using e-values provides a rather good fit to the empirical data, while in Figure 5 its match to the same empirical data appears to be substantially worse. I suspect that this is because the value difference on the x-axis in Figure 1 is based on the e-values, while in Figure 5 it is based on the r-values from the Reval algorithm. However, the computation of the value difference measure on the two x-axes is not explicitly described in the figures or methods section and these details should be added to the manuscript. If my guess is correct, then I think it is misleading to plot the DDM fit to e-values against choice and RT curves derived from r-values. Comparing Figures 1 and 5, it seems that changing the axes creates an artificial impression that the DDM using e-values is much worse than the one fit using r-values.

We agree with the reviewer that this way of presenting the DDM fits could be misleading. In the previous version of the manuscript, we included the two fits in the same figure panel to make it clear that the sensitivity (slope) of the choice function is greater when we fit the data using the r-values (now d-values) than when we fit them using the e-values (now s-values). In the revised version of Figure 5, we include the data points already shown in Figure 1, so that each DDM fit is shown with their corresponding data points. Thus we avoid giving the false impression that the DDM model fit using the s-values is much worse than the one fit using the d-values. This said, the fit is indeed worse, as we now show with the formal model comparison suggested by the reviewer (next comment).

Relatedly, do model comparison metrics favor a DDM using r-values over one using e-values in any of the datasets tested? Such tests, which use the full distribution of response times without dividing the continuum of decision difficulty into arbitrary hard and easy bins, would be more convincing than the tests of RT differences between the categorical divisions of hard versus easy.

We now include the model comparison suggested by the reviewer. The comparison shows that the DDM model using dynamic values explains the choice and response time data better than one using static values. One potential caveat of this comparison, which explains why we did not include it in the original version of the manuscript, is that the d-values are obtained from a fit to the choice data, which could bias the subsequent DDM comparison. We control for this in three ways: (1) by calculating the difference in Bayesian Information Criterion (BIC) between the models, penalizing the DDM model that uses the d-values for the additional parameter (δ); (2) by comparing the difference in BIC against simulations of a model in which the choice and RT data were obtained assuming static values; this analysis shows that if values were static, the DDM using static values would be favored in the comparison despite having one fewer parameter; (3) ignoring the DDM fit to the choices in the model comparison, and just comparing how well the two models explain the RTs; this comparison is unbiased because the δ values are fit only to the choice data, not the RTs. These analyses are now included in Figure 5 and Figure 5–Figure supplement 2.

Revaluation and reduction in the imprecision of subjective value representations during (or after) a choice are not mutually exclusive. The fact that applying Reval in the forward trial order leads to lower deviance than applying it in the backwards order (Figure 7) suggests that revaluation does occur. It doesn't tell us if there is also a reduction in imprecision. A comparison of backwards Reval versus no Reval would indicate whether there is a reduction in imprecision in addition to revaluation. Model comparison metrics and plots of the deviance from the logistic regression fit using e-values against backward and forward Reval models would be useful to show the relative improvement for both forms of Reval.

We agree with the reviewer that the occurrence of revaluation does not preclude other factors from affecting valuation. Following the reviewer’s suggestion we added a panel to Figure 6 (new panel B), in which we show the change in the deviance from the logistic regression fits between *Reval* (forward direction) and *no-Reval*. The figure clearly shows that the difference in deviance for the data is much larger than that obtained from simulations of choice data generated from the logistic fits to the static values (shown in red).

Interestingly, we also observe that the deviance obtained after applying Reval in the backward direction is lower than that obtained using the s-values. We added a panel to figure 7 showing this (Fig. 7B). This observation, however, does not imply that there are factors affecting valuation besides revaluation (e.g.,”reduction in imprecision”). Indeed, as we now show in a new panel in Figure 11 (panel F), the same effect (lower deviance for backward Reval than no-Reval) is observed in simulations of the *ceDDM*.

Besides the new figure panels (Fig. 6B, 7B, 11F), we mention in Discussion (new subsection, “Limitations...”, paragraph #2) the possibility that there are other non-dynamic contributions to the reduction in deviance for Backward Reval compared to no-Reval:

“Another limitation of our study is that, in one of the datasets we analyzed (Sepulveda et al. 2020), applying *Reval* in the forward direction was no better than applying it in the backward direction (Fig. 10). We speculate that this failure is related to idiosyncrasies of the experimental design, in particular, the use of alternating blocks of trials with different instructions (select preferred vs. select non-preferred). More importantly, *Reval* applied in the backward direction led to a significant reduction in deviance relative to that obtained using the static values. This reduction was also observed in the ceDDM, suggesting that the effect may be explained by the changes in valuation during deliberation. However, we cannot discard a contribution from other, non-dynamic changes in valuation between the rating and choice phase including contextual effects (Lichtenstein and Slovic, 2006), stochastic variability in explicit value reporting (Polania et al., 2019), and the limited range of numerical scales used to report value.”

Did the analyses of BOLD activity shown in Figure 9 orthogonalize between the various e-valueand r-value-based regressors? I assume they were not because the idea was to let the two types of regressors compete for variance, but orthogonalization is common in fMRI analyses so it would be good to clarify that this was not used in this case. Assuming no orthogonalization, the unique variance for the r-value of the chosen option in a model that also includes the e-value of the chosen option is the delta term that distinguishes the r and e-values. The delta term is a scaled count of how often the food item was chosen and rejected in previous trials. It would be useful to know if the vmPFC BOLD activity correlates directly with this count or the entire r-value (e-value + delta). That is easily tested using two additional models that include only the r-value or only the delta term for each trial.

We did not orthogonalize the static value and dynamic value regressors. We have included this detail in the revised methods. We thank the reviewer for the suggestion to run additional models to improve our ability to interpret our findings. We have substantially revised all fMRI-related sections of the paper. We took this opportunity to apply standardized and reproducible preprocessing steps implemented in *fmriprep*, present whole-brain corrected maps on a reconstructed surface of a template brain, and include links to the full statistical maps for the reader to navigate the full map, rather than rely on the static image in the figures. We implemented four models in total: model 1 includes both static value (Vs) obtained during the auction procedure prior to the choice phase and dynamic value (Vd) output by the revaluation algorithm (similar to the model presented in the first submission); model 2 includes only delta = Vd - Vs; model 3 includes only Vs; model 4 includes only Vd. All models included the same confound and nuisance regressors. We found that Vd was positively related to BOLD in vmPFC when accounting for Vs, correcting for familywise error rate at the whole brain level. Interestingly, the relationship between delta and vmPFC BOLD did not survive whole-brain correction and the effect size of the relationship between Vd and vmPFC bold in model 4 was larger than the effect size of the relationship between Vs and vmPFC bold in model 3 and survived correction at the whole brain level encompassing more of the vmPFC. Together, these findings bolster our claim that Vd better accounts for BOLD variability in vmPFC, a brain region reliably linked to valuation.

Please confirm that the correlation coefficients shown in Figure 11 B are autocorrelations in the MCMC chains at various lags. If this interpretation is incorrect, please give more detail on how these coefficients were computed and what they represent.

We added a paragraph in Methods explaining how we compute the correlations in Figure 11B (last paragraph of the sub-section “Correlated-evidence DDM” in Methods):

“The correlations in Fig. 11B were generated using the best-fitting parameters for each participant to simulate 100,000 Markov chains. We generate Markov chain samples independently for the left and right items over a 1-second period. To illustrate noise correlations, the simulations assume that the static value of both the left and right items is zero. We then and for each of the Markov chains (𝑥). Pearson's𝑥 correlation is computed between these 𝑡 calculate the difference in dynamic value (𝑥) between the left and right items at each time (𝑡) differences at time zero, 𝑥𝑖(𝑡 = 0), and at time 𝑥𝑖(𝑡 = τ), for different time lags τ. Correlations were calculated independently for each participant. Each trace in Fig. 11B represents a different participant.”

The paper presents the ceDDM as a proof-of-principle type model that can reproduce certain features of the empirical data. There are other plausible modifications to bounded evidence accumulation (BEA) models that may also reproduce these features as well or better than the ceDDM. For example, a DDM in which the starting point bias is a function of how often the two items were chosen or rejected in previous trials. My point is not that I think other BEA models would be better than the ceDDM, but rather that we don't know because the tests have not been run. Naturally, no paper can test all potential models and I am not suggesting that this paper should compare the ceDDM to other BEA processes. However, it should clearly state what we can and cannot conclude from the results it presents.

Indeed, the *ceDDM* should be interpreted as a proof-of-principle model, which shows that drifting values can explain many of our results. It is definitely wrong in the details, and we are open to the possibility that a different way of introducing sequential dependencies between decisions may lead to a better match to the experimental data. We now mention this in a new subsection of Discussion, “Limitations...” paragraph #3:

“Finally, we emphasize that the *ceDDM* should be interpreted as a proof-of-principle model used to illustrate how stochastic fluctuations in item desirability can explain many of our results. We chose to model value changes following an MCMC process. However, other stochastic processes or other ways of introducing sequential dependencies (e.g., variability in the starting point of evidence accumulation) may also explain the behavioral observations. Furthermore, there likely are other ways to induce changes in the value of items other than through past decisions. For example, attentional manipulations or other experiences (e.g., actual food consumption) may change one's preference for an item. The current version of the *ceDDM* does not allow for these influences on value, but we see no fundamental limitation to incorporating them in future instantiations of the model.”

This work has important practical implications for many studies in the decision sciences that seek to understand how various factors influence choice outcomes. By better accounting for the context-specific nature of value construction, studies can gain more precise estimates of the effects of treatments of interest on decision processes.

Thank you!

That said, there are limitations to the generalizability of these findings that should be noted.These limitations stem from the fact that the paper only analyzes choices between food items and the outcomes of the choices are not realized until the end of the study (i.e., participants do not eat the chosen item before making the next choice). This creates at least two important limitations. First, preferences over food items may be particularly sensitive to mindsets/bodily states. We don't yet know how large the choice deltas may be for other types of goods whose value is less sensitive to satiety and other dynamic bodily states. Second, the somewhat artificial situation of making numerous choices between different pairs of items without receiving or consuming anything may eliminate potential decreases in the preference for the chosen item that would occur in the wild outside the lab setting. It seems quite probable that in many real-world decisions, the value of a chosen good is reduced in future choices because the individual does not need or want multiples of that item. Naturally, this depends on the durability of the good and the time between choices. A decrease in the value of chosen goods is still an example of dynamic value construction, but I don't see how such a decrease could be produced by the ceDDM.

These are all great points. The question of how generalizable our results are to other domains is wide open. We do have preliminary evidence suggesting that in a perceptual decision-making task with two relevant dimensions (motion and color; Kang, Loffler et al. eLife 2021), the dimension that was most informative to resolve preference in the past is prioritized in future decisions. We believe that a similar process underlies the apparent change in value in value-based decisions. We decided not to include this experiment in the manuscript, as it would make the paper much longer and the experimental designs are very different. Exploring the question of generality is a matter for future studies.

We also agree that food consumption is likely to change the value of the items. For example, after eating something salty we are likely to want something to drink. We mention in the revised manuscript that time, choice deliberation, attentional allocation and other experiences (including food consumption) are likely to change the value of the alternatives and thus affect future choices and valuations.

The *ceDDM* captures only sequential dependencies that can be attributed to values that undergo diffusion-type changes during deliberation. While the ceDDM captures many of the experimental observations, the value of an item may change for reasons not captured by the *ceDDM*. For example, food consumption is likely to change the value of items (e.g., wanting something to drink after eating something salty). The reviewer is correct that the current version of *ceDDM* could not account for these changes in value. However, we see no fundamental limitation to extending the *ceDDM* to account for them.

We discuss these issues in a new subsection in Discussion (“Limitations...” paragraph #3).

**Recommendations for the authors:**

**Reviewer #1 (Recommendations For The Authors):**
SummaryThe authors address assumptions of bounded accumulation of evidence for value-based decision-making. They provide convincing evidence that subjects drift in their subjective preferences across time and demonstrate valuable methods to detect these drifts in certain task designs.My specific comments are intended to assist the authors with making the paper as clear as possible. My only major concern is with the reporting of the fMRI results.

Thank you, please see our responses above for a description of the changes we made to the fMRI analyses.

Specific comments- In the intro, I would ask the authors to consider the idea that things like slow drift in vigilance/motivation or faster drifts in spatial attention could also generate serial dependencies in perceptual tasks. I think the argument that these effects are larger in value-based tasks is reasonable, but the authors go a bit too far (in my opinion) arguing that similar effects do not exist *at all* in perceptual decision-making.

We added a sentence in the Discussion (new section on Limitations, paragraph #1) mentioning some of the literature on sequential dependencies in perceptual tasks and asking whether there might be a common explanation for such dependencies for perceptual and value-based decisions. We tried including this in the Introduction, but we thought it disrupted the flow too much.

- Figure 1: would it not be more clear to swap the order of panels A and B? Since B comes first in the task?

We agree, we swapped the order of panels A and B.

- Figure 2: the label 'simulations' might be better as 'e-value simulations'

Yes, we changed the label ‘simulations’ to ‘simulations with s-values’ (we changed the term *explicit* value to *static* value, following a suggestion by Reviewer #2).

- For the results related to Figure 2, some citations related to gaps between "stated versus revealed preferences" seem appropriate.

We added a few relevant citations where we explain the results related to Figure 2.

- Figure 3: in addition to a decrease in match preferences over the session, it would be nice to look at other features of the task which might have varied over the session. e.g. were earlier trials more likely to be predicted by e-value?

We do see a trend in this direction, but the effect is not significant. The following figure shows the consistency of the choices with the stated values, as a function of the |∆value|, for the first half (blue) and the second half (red) of the trials. The x-axis discretizes the absolute value of the difference in static value between the left and right items, binned in 17 bins of approximately equal number of trials.

The slope is shallower for the second half, but a logistic regression model revealed that the difference is not significant:,logit⁡[pconsistent]=β0+β1|Δvs|+β2|Δvs|×Ilate 

where Ilate is an indicator variable that takes a value of 1 for the second half of the trials and zero otherwise.

As expected from the figure β2 was negative (-0.15) but the effect was not significant (p-value = 0.32, likelihood ratio test).

We feel we do not have much to say about this result, which may be due to lack of statistical power, so we would rather not include this analysis in the revised manuscript.

It is worth noting that if we repeat the analysis using the dynamic values obtained from Reval instead of the static values, the consistency is overall much greater and little difference is observed between the first and second halves of the experiment:

**Author response image 2. sa3fig2:** 

- The e-value DDM fit in Figure 1C/D goes through the points pretty well, but the e-value fits in 5A do not because of a mismatch with the axis. The x-axis needs to say whether the value difference is the e-value or the r-value. Also, it seems only fair to plot the DDM for the r-value on a plot with the x-axis being the e-value.

Thank you for this comment, we have now changed Figure 5A, such that both sets of data points are shown (data grouped by both e-values and by r-values). We agree that the previous version made it seem as if the fits were worse for the DDM fit to the e-values. The fits are indeed worse, as revealed by a new DDM model comparison (Figure 5–Figure supplement 2), but the effect is more subtle than the previous version of the figure implied.

- How is Figure 5B "model free" empirical support? The fact that the r-value model gives better separation of the RTs on easy and hard trials doesn't seem "model-free" and also it isn't clear how this directly relates to being a better model. It seems that just showing a box-plot of the R2 for the RT of the two models would be better?

We agree that “model free” may not be the best expression, since the r-values (now d-values) are derived from a model (Reval). Our intention was to make clear that because Reval only depends on the choices, the relationship between RT and ∆vdynamic is a prediction. We no longer use the term, model free, in the caption. We tried to clarify the point in Results, where we explain this figure panel. We have also included a new model comparison (Figure 5–Figure supplement 2), showing that the DDM model fit to the d-values explains choice and RT better than one fit to the s-values.

This said, we do consider the separation in RTs between easy and hard trials to be a valid metric to compare the accuracy of the static and dynamic values. The key assumption is that there is a monotonically decreasing relationship between value difference, ∆v, and response time. The monotonic relationship does not need to hold for individual trials (due to the noisiness of the RTs) but should hold if one were to average a large enough number of trials for each value of ∆v.

Under this assumption, the more truthful a value representation is (i.e., the closer the value we infer is to the true subjective value of the item on a given trial, assuming one exists), the greater the difference in RTs between trials judged to be difficult and those considered easy. To illustrate this with an extreme case, if an experimenter’s valuation of the items is very inaccurate (e.g., done randomly), then on average there will be no difference between easy and difficult RTs as determined by this scoring.

- Line 189: Are the stats associated with Eq 7, was the model fit subject by subject? Combining subjects? A mixed-effects model? Why not show a scatter plot of the coefficients of Δvₑ and Δvᵣ (1 point/subject).

The model was not fit separately for each subject. Instead, we concatenated trials from all subjects, allowing each subject to have a different bias term (β0,i).

We have now replaced it with the analysis suggested by the reviewer. We fit the logistic regression model independently for each participant. The scatter plot suggested by the reviewer is shown in Figure 5–Figure supplement 1. Error bars indicate the s.e. of the regression coefficients:

It can be seen that the result is consistent with what we reported before: βd is significantly positive for all participants, while βs is not.

- I think Figure S1 should be a main figure.

Thank you for this suggestion, we have now included the former Figure S1 as an additional panel in Figure 5.

- Fig 9 figure and text (line 259) don't exactly match. In the text it says that the BOLD correlated with vᵣ and not vₑ, but the caption says there were correlations with vᵣ after controlling for vₑ. Is there really nothing in the brain that correlated with vₑ? This seems hard to believe given how correlated the two estimates are. In the methods, 8 regressors are described. A more detailed description of the results is needed.

Thank you for pointing out the inconsistency in our portrayal of the results in the main text and in the figure caption. We have substantially revised all fMRI methods, re-ran fMRI data preprocessing and implemented new, simpler, and more comprehensive GLM models following Reviewer #2's suggestion. Consequently, we have replaced Figure 9, added Figure 9 — Figure Supplement 1, and uploaded all maps to NeuroVault. These new models and maps allow for a clearer interpretation of our findings. More details about the fMRI analyses in the methods and results are included in the revision. We took care to use similar language in the main text and in the figure captions to convey the results and interpretation. The new analyses strengthen our original conclusion: dynamic values better explain BOLD activity in the ventromedial prefrontal cortex, a region consistently associated with valuation, than static values.

- It's great that the authors reanalyzed existing datasets (fig 10). I think the ΔRT plots are the least clear way to show that _reval_ is better. Why not a figure like Figure 6a and Figure 7 for the existing datasets?

We agree with the reviewer. We have replaced Fig. 10 with a more detailed version. For each dataset, we show the ΔRT plots, but we also show figures equivalent to Fig. 6a, Fig. 7a, and the new Fig. 6b (Deviance with and without *Reval*).

**Reviewer #2 (Recommendations For The Authors):**
I assume that the data and analysis code will be made publicly and openly available once the version of record is established.

Yes, the data and analysis code is now available at: https://github.com/arielzylberberg/Reval_eLife_2024

We added a Data Availability statement to the manuscript.